# Rényi complexity in mean-field disordered systems

**Nina Javerzat$^\star$, Eric Bertin, and Misaki Ozawa**

Université Grenoble Alpes, CNRS, LIPhy, 38000 Grenoble, France

$\star$ nina.javerzat@univ-grenoble-alpes.fr

## Abstract

Configurational entropy, or complexity, plays a critical role in characterizing disordered systems such as glasses, yet its measurement often requires significant computational resources. Recently, Rényi entropy, a one-parameter generalization of Shannon entropy, has gained attention across various fields of physics due to its simpler functional form, making it more practical for measurements. In this paper, we compute the Rényi version of complexity for prototypical mean-field disordered models, including the random energy model, its generalization, referred to as the random free energy model, and the $p$-spin spherical model. We first demonstrate that the Rényi complexity with index $m$ is related to the free energy difference for a generalized annealed Franz-Parisi potential with $m$ clones. Detailed calculations show that for models having one-step replica symmetry breaking (RSB), the Rényi complexity vanishes at the Kauzmann transition temperature $T_K$, irrespective of $m > 1$, while RSB solutions are required even in the liquid phase. This study strengthens the link between Rényi entropy and the physics of disordered systems and provides theoretical insights for its practical measurements.

# 1 Introduction

High-dimensional, rugged (free) energy landscapes are a hallmark of disordered systems, including structural glasses [1–3], spin glasses [4, 5], constraint satisfaction problems [6–8], machine learning models [9, 10], and biological systems [11, 12]. A key quantity in understanding these landscapes is the complexity (or configurational entropy), $\Sigma$, which quantifies the number of metastable states within the landscape, providing crucial insights into the statistical characterization of these systems [13, 14]:

$$\Sigma = \frac{1}{N} \log \mathcal{N}, \tag{1}$$

where $\mathcal{N}$ is the number of metastable state and $N$ is the number of elements (e.g., particles, spins). In this paper, the natural logarithm is used unless otherwise stated. $\Sigma$ defined in Eq. (1) is based on Boltzmann's view on entropy, i.e., the logarithm of the number of accessible states. Alternatively, particularly in thermal equilibrium, one can see $\Sigma$ based on Gibbs' view or Shannon's information-theoretical view [15] using the probability distribution, $p_\alpha$, for finding a metastable state $\alpha$:

$$\Sigma = -\frac{1}{N} \sum_\alpha p_\alpha \log p_\alpha. \tag{2}$$

$\Sigma$ has been measured numerically, or computed analytically in a wide variety of disordered systems [5].

    In structural glasses, $\Sigma$ is one of the most fundamental quantities in theories of the glass transition [16–20]. $\Sigma$ takes a finite value in a supercooled liquid, and it decreases with decreasing temperature. A sharp reduction of $\Sigma$ reflects rarefaction of the number of accessible metastable states, leading to glassy slow dynamics [18,19]. The mean-field theory of the glass transition [17, 20] predicts that when the temperature is decreased further, $\Sigma$ vanishes at a finite temperature $T_K$, called the Kauzmann transition temperature [21,22], where the system undergoes a phase transition from a supercooled liquid to an ideal glass.

    Although the complexity provides us with valuable insights into the phenomenology of glassy systems, its practical measurement involves multiple difficulties [23]. First, in finite dimensions, metastable states are no longer well-defined since the energy barrier between states is finite (unlike mean-field models). Thus, metastable states are meaningful only in a short (vibrational) timescale [24]. Second, direct (brute force) counting of metastable states is virtually impossible except for very small $N$ (say $N \approx 20$) because of an exponentially large number of states [25]. Various computational schemes have been proposed to circumvent these difficulties [26, 27] (see Ref. [23] for review). For example, the inherent structure formalism approximates the free energy landscape by a potential energy landscape at $T \to 0$

and computes the associated complexity [28–30]. Thermodynamic integration schemes were introduced by imposing a harmonic potential to confine the system in a glass state [31, 32]. Among various proposals, measuring the free energy difference between the liquid and glass states, using the so-called Franz-Parisi potential (cf. Sec. 2.4) is the most straightforward and theoretically grounded [33–35]. However, computation of the (quenched) Franz-Parisi potential requires a thermal average of the system (replica 2) under the external field coupled to a reference configuration (replica 1). One then performs averaging over independent reference configurations. This double (quenched) average requires huge computational resources [36]. Thus, in the literature, an annealed version of the Franz-Parisi potential, where two replicas evolve on the same timescale (hence, single, annealed, average), is often computed as a proxy to the quenched Franz-Parisi potential [37, 38]. Although recent developments in efficient sampling algorithms, such as swap Monte-Carlo [39], allow to perform simulations for the quenched Franz-Parisi potential, the temperature range and system size is still limited due to harsh computational costs [40–42]. Despite the frequent use of the annealed Franz-Parisi potential, its validity as an approximation of the quenched one has yet to be investigated widely (except for earlier numerical simulations [36]).

The difficulty of measuring the Shannon entropy in Eq. (2) originates from its daunting functional form, involving the probability density times *the logarithm of* the probability density. This difficulty also arises in other domains of physics: for example, in quantum many-body systems, the Shannon entropy corresponds to the von Neumann entropy, which is a measure of quantum entanglement. However, direct measurement of the von Neumann entropy is challenging in experiments, simulations, and analytical calculations.

Recently, there has been growing interest in using the Rényi entropy as an alternative to the Shannon entropy [43–53]. The Rényi entropy is a one-parameter generalization of Shannon entropy (with the parameter denoted as $m$ here), which was initially proposed in information theory [54], defined as

$$\Sigma^{\text{Renyi}}(m) = \frac{1}{N(1-m)} \log \sum_\alpha (p_\alpha)^m. \tag{3}$$

As explained in detail in Sec. 2.3, the Rényi entropy coincides with the Shannon entropy in the limit $m \to 1$. The probability density $p_\alpha$ enters the Rényi entropy in Eq. (3) in the form of $\sum_\alpha (p_\alpha)^m$, while it enters the Shannon entropy via $\sum_\alpha p_\alpha \log p_\alpha$. The former functional form is simpler than the latter in terms of measurements (in experiments and simulations) and analytical calculations. This is one of the main reasons why the Rényi entropy is widely used. In quantum many-body systems, the Rényi entropy can also serve as a measure of quantum entanglement, that can be experimentally measured [43, 44, 55] and is computationally less demanding in numerical simulations than the Shannon (or von Neumann) entanglement entropy [45–48, 56]. Moreover, computing the Rényi entropy for integer $m$, it is possible to obtain the entanglement entropy by analytic continuation [57]. A similar situation can be found in the characterization of chaos in dynamical systems: instead of a direct measurement of the Kolmogorov-Sinai entropy, which has essentially the same functional form as the Shannon one, it is convenient to compute the Rényi entropy as a proxy of the former [58–60]. The Rényi entropy also has applications in non-equilibrium statistical mechanics [49, 60–62], and is frequently used in ecology and statistics. In particular, Hill's diversity index (corresponding to the exponential of the Rényi entropy) characterizes the diversity of a statistical ensemble [50, 63]. Recently, Wang and Harrowell [51] proposed structural diversity, inspired by biodiversity literature, using the (exponential of) Rényi entropy to characterize various crystalline and amorphous materials quantitatively. Aside from these examples, Rényi entropy is applied in a wide variety of physics research areas [52, 53, 64].

However, the use of Rényi entropy is not just for practical purposes, and its relevance in physics is not a coincidence. Characterizing the mean of information content (or logarithm of

probability) is ubiquitous in physics. In Sec. 2.3, we detail the construction of Rényi entropy in terms of a generalized mean of information content possessing the additivity property.

One of the main goals of this paper is to demonstrate the role of the Rényi entropy in the context of disordered systems, along the direction initiated by Kurchan and Levine [65], which shows the deep connections between the Rényi entropy and the physics of disordered systems. Kurchan and Levine interpret the thermodynamics of the glass transition through a Rényi version of the complexity [65] (see also a short review [66]). In particular, they clarified the relationship between the Rényi complexity and the so-called Monasson approach (see Sec. 2.1). Furthermore, going beyond mean-field, they associated the Rényi complexity with frequently repeated amorphous patches in real space structure [67], proposing a practical method to estimate the Rényi complexity in finite dimensions [65]. Thus, the Rényi complexity is not just a convenient analog of the (Shannon) complexity, but it is an insightful quantity to assess fundamental aspects of the glass transition.

In this paper, we extend the phenomenological arguments put forward by Kurchan and Levine and investigate the Rényi complexity of disordered models in detail. First, in section 2.1 we review the computation of the complexity using the Monasson method. In section 2.3 we introduce the Rényi complexity as a generalization of the Shannon one, and in section 2.4 we leverage the connection between the Rényi complexity and the Monasson approach, to demonstrate that the Rényi complexity with index $m$ essentially corresponds to the free energy difference of the $m$-components annealed Franz-Parisi potential in any dimension. We then compute the Rényi complexity for several prototypical disordered models: the Random Free Energy Model [68] –which encompasses the standard Random Energy Model [69] as a specific limit– in section 3, and the $p$-spin spherical model [70] in section 4. From a technical perspective, our computation for the $p$-spin model follows the approach of Kurchan, Parisi, and Virasoro [71], using real replicas with integer $m$ (referred to as clones in our paper), but extends this to a more detailed analysis for general real values of $m$. For each model we provide detailed temperature and $m$ dependencies, in the liquid phase above the Kauzmann transition. Our computations across all models studied show that all Rényi complexities with index $m$ vanish at the same Kauzmann transition temperature, $T_K$. However, the solutions require a replica symmetry breaking ansatz, even in the liquid phase, and we provide the phase diagram for the RS/RSB transition.

Our results provide deeper insights into the Rényi complexity in disordered systems, particularly in models exhibiting one-step replica symmetry breaking. Additionally, they offer an insightful guideline for using the Rényi complexity (and the annealed Franz-Parisi potential) as an estimate of complexity in numerical simulations.

## 2 Rényi entropy and related approaches

We first give a brief pedagogical review of the Monasson approach, before explaining how it is connected to the Shannon complexity. This also sets up our notations for later use. We then detail how the Rényi complexity provides a generalization of the Shannon one. Finally we introduce the Franz-Parisi potentials and explain that the Rényi complexity corresponds to a generalized annealed Franz-Parisi potential.

### 2.1 Monasson approach for computing the complexity

In mean-field theories, a convenient way to compute the complexity $\Sigma$ is Monasson's construction [72] (see also Refs. [73–76] for reviews). Consider the partition function $Z(m)$ of a system

composed of $m$ (real number) clones belonging to the same metastable state specified by $\alpha$:

$$Z(m) = \sum_\alpha e^{-m\beta N f_\alpha(T)} = \int \mathrm{d}f \exp\left[-N\left\{m\beta f - \Sigma(f, T)\right\}\right]$$

$$\approx \exp\left[-N\left\{m\beta f_*(T, m) - \Sigma(f_*(T, m), T)\right\}\right], \tag{4}$$

where $f_\alpha(T)$ is the free energy density of the metastable state $\alpha$, and $\beta = 1/T$ is the inverse temperature. $\Sigma(f, T)$ is given by
$\Sigma(f, T) = \frac{1}{N} \log \mathcal{N}(f, T)$ with $\mathcal{N}(f, T) = \sum_\alpha \delta(f - f_\alpha(T))$, where $\delta(x)$ is the Dirac delta function. In the last equality, we performed the saddle-point approximation, so that $f_*(T, m)$ is given by the saddle-point condition,

$$m\beta = \partial \Sigma(f, T)/\partial f|_{f=f_*(T,m)}. \tag{5}$$

The free energy *per element*, $\phi(m)$ is given by

$$\beta \phi(m) = -\frac{1}{mN} \log Z(m) \approx \beta f_*(T, m) - \frac{1}{m}\Sigma(f_*(T, m), T). \tag{6}$$

The differentiation of $\phi(m)$ with respect to $m$ nicely decomposes the two contributions in Eq. (6) as

$$\Sigma(f_*(T, m), T) = m^2 \frac{\partial}{\partial m} \beta \phi(m), \tag{7}$$

$$f_*(T, m) = \frac{\partial}{\partial m} m\phi(m), \tag{8}$$

where we have used Eq. (5). In particular, Eq. (7) allows one to extract the complexity, $\Sigma$, above $T_K$ by taking the $m \to 1$ limit,

$$\Sigma = \lim_{m \to 1} m^2 \frac{\partial}{\partial m} \beta \phi(m). \tag{9}$$

Below $T_K$ instead, $m_* < 1$ is chosen such that $\Sigma(f_*(T, m_*), T) = 0$.

Thus, the computation of $\Sigma$ boils down to the computation of $\beta \phi(m)$, which is the free energy of the system composed of $m$ clones belonging to the same metastable glass state. In practice, to compute $\beta \phi(m)$ from the microscopic Hamiltonian $E_i$, where $i$ specifies configuration, one needs to constrain $m$ clones in the same state. This is realized, for instance, by introducing an overlap $q$ and computing the free energy and associated partition function in a constrained ensemble [73], as given by

$$\beta \phi(m, q) = -\frac{1}{mN} \overline{\log Z(m, q)}, \tag{10}$$

$$Z(m, q) = \sum_{i_1} \sum_{i_2} \cdots \sum_{i_m} e^{-\beta\left(E_{i_1} + E_{i_2} + \cdots + E_{i_m}\right)} \prod_{a<b}^{m} \delta(q - \hat{q}_{i_a, i_b}), \tag{11}$$

where $\hat{q}_{i_a, i_b}$ is an overlap function characterizing similarity between configurations $i_a$ and $i_b$. When $i_a$ and $i_b$ are similar $\hat{q}_{i_a, i_b} \approx 1$, whereas $\hat{q}_{i_a, i_b} \approx 0$ when $i_a$ and $i_b$ are distinct. $\overline{\cdots}$ denotes a disordered average if needed, e.g., in cases of spin glasses with disordered couplings. Ideally, we wish to compute $\beta \phi(m, q)$ for a given disorder, as the above argument stands on such a situation. Yet, in practice, thanks to the self-averaging property, at the thermodynamic limit, one can equivalently obtain $\beta \phi(m, q)$ by Eq. (10) with the disordered average. To extract $\Sigma$ at $m \to 1$, one can use $\log Z(m, q)$ instead of $\overline{\log Z(m, q)}$ for simple models such as the $p$-spin model. Yet this is not generally correct [76]. When $m \neq 1$, the distinction between

$\log \overline{Z(m,q)}$ and $\overline{\log Z(m,q)}$ is crucial, even for the $p$-spin model, as the latter must be used for the Monasson approach and the computation of Rényi complexity, while the former is related to the large deviation function of the free energy [77–80] (see further discussions below). In systems with continuous variables, such as spherical models, the summations in Eq. (11) are replaced by integrals.

In practice, to constrain $m$ clones in a metastable glass state at a given temperature $T$, we set the prescribed overlap $q$ to the finite value $q = q_{EA}(T)$, where $q_{EA}(T)$ (often satisfying $q_{EA} \approx 1$) is the Edwards-Anderson parameter at temperature $T$. Using Eqs. (9) and (10), one then extracts the complexity as $\Sigma(T) = \Sigma(q_{EA}(T))$, with

$$\Sigma(q_{EA}) = \lim_{m \to 1} m^2 \frac{\partial}{\partial m} \beta \phi(m, q_{EA}). \tag{12}$$

## 2.2 Shannon expression of the complexity

The Monasson construction allows one to compute the complexity in Shannon's information view based on the probability distribution, as given by Eq. (2). Indeed, consider the partition function of the original system with $m = 1$, $Z(m = 1) = \sum_\alpha Z_\alpha$, where $Z_\alpha = e^{-\beta N f_\alpha(T)}$ is the partition function restricted to a metastable state $\alpha$. The probability distribution $p_\alpha$ for finding a metastable state $\alpha$ can be defined as

$$p_\alpha = \frac{Z_\alpha}{Z(m = 1)} = \frac{e^{-\beta N f_\alpha(T)}}{Z(m = 1)}. \tag{13}$$

Then,

$$\begin{aligned}
-\frac{1}{N} \overline{\sum_\alpha p_\alpha \log p_\alpha} &= \beta \overline{\sum_\alpha p_\alpha f_\alpha(T)} + \frac{1}{N} \overline{\log Z(m = 1)} = \beta f_*(T, m = 1) - \beta \phi(m = 1) \\
&= \lim_{m \to 1} \Sigma(f_*(T, m), T). \tag{14}
\end{aligned}$$

We used $\overline{\sum_\alpha p_\alpha f_\alpha(T)} = f_*(T, m = 1)$, $\beta \phi(m = 1) = -\frac{1}{N} \overline{\log Z(m = 1)}$, and Eq. (6). Equation (14) is nothing but the complexity computed with Monasson's method. We then conclude

$$\Sigma = -\frac{1}{N} \overline{\sum_\alpha p_\alpha \log p_\alpha} = \frac{1}{N} \overline{\sum_\alpha p_\alpha I_\alpha}, \tag{15}$$

where $I_\alpha = -\log p_\alpha$ is the information content or magnitude of surprise. This equation provides us with Shannon's information-theoretical view on the complexity. If one observes a metastable state with a very small probability $p_\alpha$, one gets a surprise, and hence, it is informative. Instead, if one observes a state with high $p_\alpha$, it would not be surprising and not informative, because it takes place very often. Thus, $\Sigma$ based on Eq. (15) quantifies a *mean* of the information content. It also quantifies the magnitude of uncertainty on *average*. At higher temperatures, it is uncertain which state one observes among an exponentially large number of metastable states; hence, the *mean* information content $\Sigma$ is large. Instead, at lower temperatures, in particular below $T_K$, one would always observe the system in the unique stable state (actually a subexponential number of stable states). Hence, the *mean* information content is zero.

## 2.3 Rényi complexity

Given the above considerations, the Rényi entropy corresponds to a one-parameter generalization of the Shannon entropy, whose construction, motivation, and interpretation can be understood by considering two key aspects (such details on the Rényi entropy are discussed in a recent review [81].)

- *Generalized mean of information content*. As we emphasized above, the Shannon entropy is nothing but a mean value of the information content $I_\alpha$, using the standard *arithmetic mean* (or linear average). However, the notion of *mean* is not limited to *arithmetic mean*. In fact, there are various other types of *means*, such as the geometric mean, harmonic mean, and root mean square (non-linear averages). For example, one encounters the harmonic mean when computing the equivalent resistance of parallel electrical circuits. Recall also that the root mean square is one of the most used means to analyze data in science and technology. Kolmogorov and Nagumo generalized the concept of the *mean* further using a wider class of functional forms [82, 83], which allows one to define a more general measure of averaged information content.

- *Additivity*. One can then define a generalized entropy using a *generalized mean*. However, as a quantity of information, one wishes to have an entropy with the property of additivity for independent events, namely, if two random variables $A$ and $B$ are independent, an entropy $S(A, B)$ of their joint distribution is the sum of their individual entropies, $S(A, B) = S(A) + S(B)$. This is called *additivity* which serves as a fundamental (desired) property in information theory. The additivity is also naturally expected for the thermodynamic entropy in physical systems. We also note that in contrast to the Rényi entropy, other generalized entropies, such as the Tsallis entropy, do not satisfy additivity [84–87].

Alfréd Rényi searched for a generalized entropy using the concept of generalized mean, while keeping the additivity condition, and obtained the entropy that is now called the Rényi entropy [54]. In our current setting (with the disordered average), we now define the Rényi complexity, $\Sigma^{\text{Renyi}}(m)$, by

$$\Sigma^{\text{Renyi}}(m) = \frac{1}{N(1-m)} \overline{\log \sum_\alpha (p_\alpha)^m}, \tag{16}$$

where $m$ is the Rényi index with $0 < m < \infty$ and $m \neq 1$. The Renyi complexities, $\Sigma^{\text{Renyi}}(0)$, $\Sigma^{\text{Renyi}}(1)$, and $\Sigma^{\text{Renyi}}(\infty)$ are defined as the corresponding limits of $\Sigma^{\text{Renyi}}(m)$ for $m \to 0$, $m \to 1$, and $m \to \infty$, respectively. In this study, we mainly consider $m > 1$.

The $m \to 1$ limit corresponds to the (Shannon) complexity, as one can easily check using l'Hôpital's rule:

$$
\begin{aligned}
\lim_{m \to 1} \Sigma^{\text{Renyi}}(m) &= \lim_{m \to 1} \frac{1}{N(1-m)} \overline{\log \sum_\alpha (p_\alpha)^m} = \lim_{m \to 1} \frac{\overline{\frac{\partial}{\partial m} \log \sum_\alpha (p_\alpha)^m}}{-N} \\
&= \lim_{m \to 1} \frac{\overline{\frac{1}{\sum_\alpha (p_\alpha)^m} \sum_\alpha \frac{\partial}{\partial m} e^{m \log p_\alpha}}}{-N} = -\frac{1}{N} \overline{\sum_\alpha p_\alpha \log p_\alpha} = \Sigma.
\end{aligned} \tag{17}
$$

The $m \to \infty$ limit corresponds to the so-called min-entropy, $\Sigma^{\text{Renyi}}_\infty = \lim_{m \to \infty} \Sigma^{\text{Renyi}}(m)$. When $m$ is very large, the state with the highest propbability, $\max_\alpha \{p_\alpha\}$, dominates the summation, i.e., $\sum_\alpha (p_\alpha)^m \approx (\max_\alpha \{p_\alpha\})^m$. Thus at $m \to \infty$, one obtains

$$\Sigma^{\text{Renyi}}_\infty = -\frac{1}{N} \overline{\log \max_\alpha \{p_\alpha\}} = \frac{1}{N} \overline{\min_\alpha \{-\log p_\alpha\}}. \tag{18}$$

In particular, using Eq. (13), one gets

$$\Sigma^{\text{Renyi}}_\infty = \beta \overline{f_{\text{L}}(T)} - \beta \phi(m=1), \tag{19}$$

where $f_{\text{L}}(T) = \min_\alpha \{f_\alpha(T)\}$ is the lowest free energy at a given temperature $T$.

$\Sigma^{\text{Renyi}}(m)$ is a non-increasing function as one can check $\frac{\partial \Sigma^{\text{Renyi}}(m)}{\partial m} \leq 0$. Thus, $\Sigma^{\text{Renyi}}(m)$ is bounded by $\Sigma^{\text{Renyi}}_\infty$ from below, i.e., $\Sigma^{\text{Renyi}}_\infty \leq \Sigma^{\text{Renyi}}(m)$. For $m > 1$, one can also obtain an upper bound. In general, $\sum_\alpha (p_\alpha)^m \geq \max_\alpha\{ (p_\alpha)^m\} = (\max_\alpha\{p_\alpha\})^m$. Using the definition of the Rényi entropy in Eq. (16) and min-entropy in Eq. (18), we get $\Sigma^{\text{Renyi}}(m) \leq \frac{m}{m-1} \Sigma^{\text{Renyi}}_\infty$. To summarize, when $m > 1$, we have

$$\Sigma^{\text{Renyi}}_\infty \leq \Sigma^{\text{Renyi}}(m) \leq \frac{m}{m-1} \Sigma^{\text{Renyi}}_\infty. \tag{20}$$

As it is clear from the derivation, the upper bound is realized when the summation is completely dominated by the state with the highest probability or lowest free energy. In general, a larger value of $m$ in $\Sigma^{\text{Renyi}}(m)$ tends to discriminate or highlight states with larger probability, while a smaller value of $m$ tends to take into account states with finite probabilities in a rather equal manner. Thus, varying $m$ from the Shannon entropy limit $m \to 1$ corresponds to biasing ($m > 1$) or unbiasing ($m < 1$) the original probability distribution.

## 2.4 Franz-Parisi potentials

We now explain the relation between the Rényi complexities and the Franz-Parisi potentials. The Franz-Parisi potential, $V(q)$ [33, 34, 88], corresponds to the Landau free energy for the glass transition. It is a function of the order parameter $q$ associated with the Kauzmann ideal glass transition, namely the overlap function that we introduced in section 2.1. According to mean-field theories, at high temperatures, $V(q)$ shows a single minimum near $q \approx 0$, which corresponds to the liquid state. Below Mode-Coupling dynamic transition temperature, $V(q)$ develops a second minimum at a finite overlap (the Edwards-Anderson parameter) $q = q_{\text{EA}} \approx 1$ corresponding to the metastable glass state. The second minimum (hence $V(q_{\text{EA}})$) decreases with decreasing temperature and coincides with $V(q \approx 0)$ (which is often set to zero) at $T_K$, showing a first-order-transition-like behavior. Yet, the complexity $\Sigma$ remains continuous without latent heat. Therefore, this type of transition is unique to disordered glassy systems and is called "random first-order transition" (RFOT) [18, 19]. The free energy difference between the liquid and glass states amounts to the complexity (times temperature) [33], namely,

$$\Sigma = \beta \left[ V(q_{\text{EA}}) - V(q \approx 0) \right]. \tag{21}$$

This equation provides an alternative interpretation of the complexity as a free energy difference in the Franz-Parisi potential. The random-first-order transition scenario is not restricted to structural glasses. Similar phenomenologies are observed in a wide variety of problems. In fact, the original RFOT argument was constructed based on a class of mean-field spin glasses showing one-step replica symmetry breaking [89, 90].

Franz-Parisi potentials can be defined by the quenched way, denoted as $\beta V^{\text{Quench}}(q)$ and the annealed way, denoted as $\beta V^{\text{Anneal}}(q)$:

$$\beta V^{\text{Quench}}(q) = -\frac{1}{N} \overline{\sum_{i_1} \frac{e^{-\beta E_{i_1}}}{Z} \log \sum_{i_2} \frac{e^{-\beta E_{i_2}}}{Z} \delta(q - \hat{q}_{i_1,i_2})}, \tag{22}$$

$$\beta V^{\text{Anneal}}(q) = -\frac{1}{N} \log \overline{\sum_{i_1} \sum_{i_2} \frac{e^{-\beta\left(E_{i_1} + E_{i_2}\right)}}{(Z)^2} \delta(q - \hat{q}_{i_1,i_2})}. \tag{23}$$

In the quenched construction, firstly, equilibrium configuration $i_1$ is sampled by $e^{-\beta E_{i_1}}/Z$, which serves as a reference or template configuration. On top of that, the target configuration $i_2$ is sampled according to $e^{-\beta E_{i_2}}/Z$, while the configuration $i_1$ is fixed permanently or quenched, together with measuring overlap $\hat{q}_{i_1,i_2}$ between $i_1$ and $i_2$. Thus, it requires a double

average (for a given disorder if it exists), which is computationally demanding in practice. We note that, in general, the sampling temperatures for $i_1$ and $i_2$ can be different, and this difference was exploited in some cases [41, 42]. Yet we consider that both temperatures are the same for simplicity in this paper.

In the annealed construction, instead, both configurations, $i_1$ and $i_2$, are sampled at the same time. In other words, two clones $i_1$ and $i_2$ evolve with the same timescale. Therefore, it has only one average operation (again, for a given disorder), reducing the computational cost significantly compared with the quenched construction. Indeed, previous literature performed molecular simulations under the annealed construction and measured $\beta V^{\text{Anneal}}(q)$ to get a proxy of $\beta V^{\text{Quench}}(q)$ [37, 38].

In the annealed construction above, we considered two clones evolving on the same timescale. One can generalize this setting to the $m$ clones setting. Namely, one can define the $m$-annealed Franz-Parisi potential, $\beta V^{\text{Anneal}}(m, q)$, given by

$$\beta V^{\text{Anneal}}(m, q) = -\frac{1}{N}\log \overline{\sum_{i_1}\sum_{i_2}\cdots\sum_{i_m}\frac{e^{-\beta\left(E_{i_1}+E_{i_2}+\cdots+E_{i_m}\right)}}{(Z)^m}\prod_{a<b}^{m}\delta(q-\hat{q}_{i_a,i_b})}.$$ (24)

One notices that this $m$-clones setting is conceptually very similar to the Monasson construction with $m$ clones (see Sec. 2.1). Indeed, with Eqs. (10) and (11), one can rewrite Eq. (24) as

$$\beta V^{\text{Anneal}}(m, q) = m\left[\beta\phi(m, q) - \beta\phi(m = 1)\right].$$ (25)

Thus, $\beta V^{\text{Anneal}}(m, q)$ is expressed by the difference of the free energy per element for the $m$-clones and the original (single) system.

## 2.5 Rényi complexity and $m$-annealed Franz-Parisi potential

Now we are in a position to derive the connection between the Rényi complexity defined in Eq. (16) and the $m$-annealed Franz-Parisi potential defined in Eq. (24). We wish to compute $\Sigma^{\text{Renyi}}(m, q)$ with a constraint by the overlap $q$, similar to the argument in Eq. (12). Using Eqs. (16) and (13) with the constraint, we get

$$\begin{aligned}\Sigma^{\text{Renyi}}(m, q) &= \frac{1}{N(1-m)}\left[\overline{\log Z(m, q)} - m\overline{\log Z(m = 1)}\right] \\ &= \frac{m}{m-1}\left[\beta\phi(m, q) - \beta\phi(m = 1)\right].\end{aligned}$$ (26)

Thus, $\Sigma^{\text{Renyi}}(m, q)$ is expressed by the difference of the free energy per element for the $m$-clones and the original (single) system. Namely, Eq. (26) demonstrates the relationship between the Rényi complexity and the Monasson approach as clarified by Ref. [65]. In particular, it is now clear that the Rényi index is nothing but the number of clones in the Monasson approach.

By comparing Eq. (25) and (26), we arrive at

$$\Sigma^{\text{Renyi}}(m, q) = \frac{1}{m-1}\beta V^{\text{Anneal}}(m, q).$$ (27)

To conclude, the Rényi complexity with the index $m$ corresponds to the $m$-annealed Franz-Parisi potential with a factor $1/(m-1)$.

In the following sections, we will compute the Rényi complexity in detail for prototypical mean-field disordered models, the Random Free Energy Model and the $p$-spin spherical model.

# 3 Random Free Energy Model

## 3.1 Definition of the model

As a simple example to illustrate the evaluation of the Rényi complexity, we first consider a slight generalization of the Random Energy Model in which the energies of the different configurations are interpreted as the free energies $F_\alpha = N f_\alpha$ of metastable states, where $N$ is the underlying number of degrees of freedom (that are not described explicitly). We call this model Random Free Energy Model (RFEM), taking inspiration from the Random Energy Random Entropy Model of Ref. [68]. The case of the standard Random Energy Model (REM) [69] is a specific limit of the RFEM, as will appear clearly below. The reason we study the RFEM instead of the standard REM is that the RFEM provides clear distinctions among total entropy $s_{\text{tot}}$, complexity $\Sigma$, and vibrational entropy $s_{\text{vib}}$ (or glass entropy for a metastable state). This makes it a useful model for illustrating the essence of the Monasson approach and for computing the Rényi complexities, while analytically tractable. In contrast, the standard REM has total entropy that is entirely configurational, i.e., $s_{\text{tot}} = \Sigma$, lacking any vibrational entropy component.

The free energy distribution $\rho(f)$ from which the free energy densities $f_\alpha$ are randomly drawn is now a Gaussian distribution with a temperature-dependent variance $J^2(T)/N$,

$$\rho(f) = \sqrt{\frac{N}{2\pi J^2(T)}} \exp\left(-\frac{N(f - f_0)^2}{2 J^2(T)}\right). \tag{28}$$

We denote as $M_N$ the total number of metastable states for a system of size $N$, and we assume that $M_N$ grows exponentially with $N$, as $M_N \sim e^{\lambda N}$. For the sake of simplicity, the parameter $\lambda$ is assumed to be temperature-independent. It should not be confused with the complexity $\Sigma$, which takes into account the probability weight $\propto e^{-\beta N f_\alpha}$ of the different metastable states. In the usual REM [69], configurations implicitly correspond to $2^N$ spin configurations as in an Ising spin model, so that $\lambda = \log 2$. In the RFEM, the sum is over metastable states which contain part of the system entropy as vibrational entropy, so that $0 < \lambda < \log 2$. The value of $\lambda$ will be determined below.

To get some insights on the temperature dependence of the parameter $J(T)$, we focus for simplicity on the RFEM derived from the Random Energy Random Entropy Model [68]. In this simple model, metastable states are assumed to have both a random energy density $\varepsilon_\alpha$ and a random entropy density $s_\alpha$ drawn from Gaussian distributions, so that $f_\alpha = \varepsilon_\alpha - T s_\alpha$. Both $\varepsilon_\alpha$ and $s_\alpha$ are temperature-independent. We assume $\overline{\varepsilon} = 0$, $\overline{\varepsilon^2} = J_0^2/N$, $\overline{s} = s_0$ and $\text{Var}(s) = \overline{s^2} - s_0^2 = \sigma^2/N$, where $J_0$ and $\sigma$ are two temperature-independent parameters. The distribution of $f_\alpha$ is thus the Gaussian distribution in Eq. (28), with

$$f_0 = -T s_0, \qquad J^2(T) = J_0^2 + T^2 \sigma^2. \tag{29}$$

The calculations performed below can in principle be done keeping $J(T)$ as a generic increasing function of $T$. However, to get tractable explicit expressions for physical observables like the typical free energy or the complexity, it is convenient to use the specific parametrization of $J(T)$ in Eq. (29). In the following, we thus keep the notation $J(T)$ as long as expressions remains simple with a generic $J(T)$, and then switch to the specific expression given in Eq. (29).

## 3.2 Thermodynamic free energy and entropy

To evaluate the thermodynamic free energy and entropy of the system, we introduce the partition function of the model, given by

$$Z = \sum_{\alpha=1}^{M_N} e^{-\beta N f_\alpha}, \tag{30}$$

The (total) thermodynamic free-energy density, $f_{\text{tot}}$, averaged over the disorder is then given by $f_{\text{tot}} = -TN^{-1}\overline{\log Z}$. The disorder-averaged quantity $\overline{\log Z}$ in the REM may be evaluated using the replica trick [91]. However, a standard approach when considering REM-type models is to introduce the density of states of a typical sample [69]. We define the density $n(f)$ of metastable states with free energy $f$. Averaging over disorder, we have for large $N$

$$\overline{n}(f) \sim e^{N[\lambda - (f-f_0)^2/2J^2(T)]}. \tag{31}$$

The average density of states is exponentially large in $N$ over the interval $f_{\text{min}} < f < f_{\text{max}}$, where

$$f_{\text{min}} = f_0 - \sqrt{2\lambda}J(T), \qquad f_{\text{max}} = f_0 + \sqrt{2\lambda}J(T). \tag{32}$$

Outside this interval, the average density of states $\overline{n}(f)$ is exponentially small in $N$, meaning that in a typical sample of the disorder in a large system, there are no states outside the interval $[f_{\text{min}}, f_{\text{max}}]$. The density of state $n_{\text{typ}}$ of a typical sample can thus be approximated as $n_{\text{typ}} \approx \overline{n}(f)$ for $f \in [f_{\text{min}}, f_{\text{max}}]$ and $n_{\text{typ}} = 0$ for $f \notin (f_{\text{min}}, f_{\text{max}})$. The partition function of a typical sample can thus be evaluated as

$$Z_{\text{typ}} \approx \int_{f_{\text{min}}}^{f_{\text{max}}} df \, n_{\text{typ}}(f) e^{-Nf/T} \approx \int_{f_{\text{min}}}^{f_{\text{max}}} df \, e^{-Ng(f)}, \tag{33}$$

where we have introduced the function

$$g(f) = -\lambda + \frac{(f-f_0)^2}{2J^2(T)} + \frac{f}{T}. \tag{34}$$

We then perform the usual approximation, $f_{\text{tot}} \approx -TN^{-1}\log Z_{\text{typ}}$.

From Eq. (33), $Z_{\text{typ}}$ can be evaluated by a saddle-point calculation. The value $f_*$ that minimizes $g(f)$ over the entire real axis is given by

$$f_* = f_0 - \frac{J^2(T)}{T}, \tag{35}$$

leading to

$$g(f_*) = -\lambda + \frac{f_0}{T} - \frac{J^2(T)}{2T^2}. \tag{36}$$

We now need to compare $f_*$ with $f_{\text{min}}$. When $f_* > f_{\text{min}}$, the typical free energy density $f_{\text{tot}} = -\frac{T}{N}\log Z_{\text{typ}}$ is obtained from the saddle-point calculation, $f_{\text{tot}} = Tg(f_*)$. Using Eqs. (32) and (35), the condition $f_* > f_{\text{min}}$ is equivalent to $T > J(T)/\sqrt{2\lambda}$. To get an explicit condition on the temperature, we use the parametrization of $J(T)$ in Eq. (29). We then find that the condition $f_* > f_{\text{min}}$ boils down to $T > T_K$, where $T_K = J_0/\sqrt{2\lambda - \sigma^2}$, provided that $\lambda > \sigma^2/2$, a condition assumed to hold in the following. In contrast, when $f_* < f_{\text{min}}$, corresponding to the low-temperature regime $T < T_K$, the free energy is given by the contribution of the lower bound of the integral, $f_{\text{tot}} = Tg(f_{\text{min}})$. It is also useful to compute the total thermodynamic entropy, $s_{\text{tot}} = -\partial f_{\text{tot}}/\partial T$, from the knowledge of the free energy $f_{\text{tot}}$. For $T > T_K$, one finds

$$f_{\text{tot}} = -T\left(\lambda + \frac{\sigma^2}{2}\right) - \frac{J_0^2}{2T} + f_0, \qquad s_{\text{tot}} = \lambda + \frac{\sigma^2}{2} - \frac{J_0^2}{2T^2} + s_0. \tag{37}$$

It follows that for $T \to \infty$, the total thermodynamic entropy density goes to $s_\infty = \lambda + \frac{\sigma^2}{2} + s_0$. Assuming that the RFEM effectively describes a spin model with $2^N$ spin configurations, one has $s_\infty = \log 2$, which fixes the parameter $\lambda$ to the value,

$$\lambda = \log 2 - s_0 - \frac{\sigma^2}{2}. \tag{38}$$

The condition $\lambda > \sigma^2/2$ then imposes the constraint $\sigma^2 < \log 2 - s_0$, which also fixes the range of $s_0$ to $0 < s_0 < \log 2$. In the following, we assume that the condition $\sigma^2 < \log 2 - s_0$ is satisfied[1]. Note that the REM case corresponds to $s_0 = 0$ and $\sigma = 0$ (i.e., metastable states boil down to single configurations with zero glass –or vibrational– entropy), and one recovers the result $\lambda_{\text{REM}} = \log 2$. For $\sigma^2 < \log 2 - s_0$, the glass transition temperature $T_K$ thus reads

$$T_K = \frac{J_0}{\sqrt{2(\log 2 - s_0 - \sigma^2)}}. \tag{39}$$

For $T < T_K$, the free energy $f_{\text{tot}}$ and thermodynamic entropy $s_{\text{tot}}$ read as

$$f_{\text{tot}} = f_0 - \sqrt{(2\log 2 - 2s_0 - \sigma^2)(J_0^2 + T^2\sigma^2)}, \qquad s_{\text{tot}} = s_0 + \sigma^2 T \sqrt{\frac{2\log 2 - 2s_0 - \sigma^2}{J_0^2 + T^2\sigma^2}}. \tag{40}$$

## 3.3 Complexity

We now evaluate the complexity counting the exponential number of metastable states at temperature $T$. To perform this calculation, we follow Monasson's approach [72], as recalled in Sec. 2.1. We thus introduce the partition function $Z(m)$ of $m$ clones,

$$Z(m) = \sum_{\alpha=1}^{M_N} e^{-m\beta N f_\alpha}, \tag{41}$$

as well as the corresponding $m$-clone free energy,

$$\phi(m) = -\frac{1}{m\beta N} \overline{\log Z(m)}. \tag{42}$$

In practice, we replace $\overline{\log Z(m)}$ in the definition of $\phi(m)$ by $\log Z_{\text{typ}}(m)$ defined as

$$Z_{\text{typ}}(m) = \int_{f_{\text{min}}}^{f_{\text{max}}} df \, e^{-N g(m,f)}, \tag{43}$$

with

$$g(m,f) = -\log 2 + s_0 + \frac{\sigma^2}{2} + \frac{(f-f_0)^2}{2J^2(T)} + \frac{mf}{T}. \tag{44}$$

Following the same reasoning as in Sec. 3.2, the value $f_*(m)$ minimizing $g(m,f)$ reads

$$f_*(m) = f_0 - \frac{mJ^2(T)}{T}. \tag{45}$$

Using Eq. (29) and assuming $\sigma^2 < 2(\log 2 - s_0)/(1+m^2)$, the condition $f_*(m) > f_{\text{min}}$ is equivalent to $T > T_c(m)$, with

$$T_c(m) = \frac{J_0}{\sqrt{\frac{2\log 2 - 2s_0 - \sigma^2}{m^2} - \sigma^2}}. \tag{46}$$

---

[1]Note that for $\log 2 - s_0 \le \sigma^2 < 2\log 2 - 2s_0$ (the upper bound corresponds to the condition $\lambda > 0$), the glass transition temperature is infinite and the model is glassy at all temperature.

We note that $T_c(m = 1) = T_K$, where $T_K$ is the Kauzmann transition temperature defined in Sec. 3.2.

One then finds for the $m$-clone free-energy,

$$
\phi(m) = 
\begin{cases}
T/m \, g(m, f_{\min}) = -(2\log 2 - 2s_0 - \sigma^2)^{1/2} J(T) + f_0, & \text{if } T < T_c(m) \\[2ex]
T/m \, g(m, f_*(m)) = -\frac{T}{2m}(2\log 2 - 2s_0 - \sigma^2) - \frac{m}{2T} J^2(T) + f_0 & \text{if } T > T_c(m).
\end{cases}
\tag{47}
$$

In contrast, if $\sigma^2 \geq 2(\log 2 - s_0)/(1 + m^2)$, or equivalently, $m \geq \sqrt{(2\log 2 - 2s_0 - \sigma^2)}/\sigma$, the condition $f_*(m) < f_{\min}$ is satisfied for all temperature, meaning that $T_c(m)$ is actually infinite. In the following, we focus on the case when $T_c(m)$ is finite, but the results straightforwardly generalize to the case of an infinite $T_c(m)$.

According to Eq. (9), the configurational entropy $\Sigma$ is obtained from $\phi(m)$ as $\Sigma = \beta \phi'(1)$. We thus get,

$$
\Sigma = 
\begin{cases}
0 & \text{if } T < T_K \\[2ex]
\log 2 - s_0 - \sigma^2 - \frac{J_0^2}{2T^2} & \text{if } T > T_K.
\end{cases}
\tag{48}
$$

One can check that $\Sigma \to 0$ when $T \to T_K^+$.

One may also evaluate the vibrational entropy, $s_{\mathrm{vib}} = s_{\mathrm{tot}} - \Sigma$. Using Eqs. (37), (40), and (48), one obtains

$$
s_{\mathrm{vib}} = 
\begin{cases}
s_0 + \sigma^2 T \sqrt{\dfrac{2\log 2 - 2s_0 - \sigma^2}{J_0^2 + T^2\sigma^2}} & \text{if } T < T_K \\[3ex]
s_0 + \sigma^2 & \text{if } T > T_K.
\end{cases}
\tag{49}
$$

One thus has a nonzero vibrational entropy in the glassy phase, which goes to zero when $T \to 0$. Note also that $s_{\mathrm{vib}}$ is continuous at $T_K$.

## 3.4 Rényi complexity

We now finally evaluate the Rényi complexity as

$$
\Sigma^{\mathrm{Renyi}}(m, T) = \frac{m\beta}{m-1} [\phi(m) - \phi(1)],
\tag{50}
$$

where $m > 1$ is now a fixed parameter. The Rényi complexity can readily be evaluated using Eqs. (47). As $T_c(m) > T_K$ for $m > 1$, three different temperature regimes have to be distinguished, namely $T < T_K$, $T_K < T < T_c(m)$, and $T_c(m) < T$. One finds

$$
\Sigma^{\mathrm{Renyi}}(m, T) = 
\begin{cases}
0 & \text{if } \quad T \leq T_K, \\[2ex]
\dfrac{mJ_0^4}{2(m-1)T^2} \left( \dfrac{1 + T/T_K}{J(T) + T\sqrt{2\log 2 - 2s_0 - \sigma^2}} \right)^2 \left(1 - \dfrac{T}{T_K}\right)^2 & \text{if } \quad T_K < T \leq T_c(m), \\[3ex]
\log 2 - s_0 - (1 + m)\dfrac{\sigma^2}{2} - \dfrac{mJ_0^2}{2T^2} & \text{if } \quad T_c(m) < T.
\end{cases}
\tag{51}
$$

We plot the obtained $\Sigma^{\mathrm{Renyi}}(m, T)$ in Fig. 1a, for $s_0 \to 0$ and $\sigma = \sqrt{\log 2}/2$. For $m \to 1$, $\Sigma^{\mathrm{Renyi}}(m, T)$ converges to the standard complexity $\Sigma(T)$ evaluated in Sec. 3.3. $\Sigma(T)$ monotonically decreases with a concave manner as the temperature is decreased and vanishes at the Kauzmann transition temperature $T_K > 0$, which is a well-known behavior. When $m$ is

increased from 1, $\Sigma^{\text{Renyi}}(m, T)$ decreases at a given $T$, which is a general property of the Rényi entropy [81]. We note that, in the regime $T_K < T \leq T_c(m)$, the expression of $\Sigma^{\text{Renyi}}(m, T)$ contains the solution associated with one step replica symmetry breaking, despite the fact that the system is above $T_K$ (this will become clear in the $p$-spin spherical model in Sec. 4). We thus plot $\Sigma^{\text{Renyi}}(m, T)$ in this intermediate regime by the solid curves. The temperature dependence of $\Sigma^{\text{Renyi}}(m, T)$ has interesting behaviors. It becomes milder with increasing $m$. In particular, the concavity seen as $m \to 1$ turns into convex behavior. Nevertheless, $\Sigma^{\text{Renyi}}(m, T)$ vanishes at the same $T_K$ irrespective of $m$. These results suggest that an accurate measurement of $\Sigma^{\text{Renyi}}(m, T)$ can provide a good estimate of the location of $T_K$.

Moreover, for arbitrary $m > 1$, the $m-$dependence factorizes in the regime, $T_K < T \leq T_c(m)$, so that:

$$\Sigma^{\text{Renyi}}(m, T) = \frac{m}{m-1} \Sigma_\infty^{\text{Renyi}}(T), \quad \forall m > 1, \quad T_K < T \leq T_c(m), \tag{52}$$

where $\Sigma_\infty^{\text{Renyi}}(T)$ is the min-entropy defined in section 2.3. As shown in Eq. (20), $\Sigma_\infty^{\text{Renyi}}(T)$ provides us with lower and upper bounds on $\Sigma^{\text{Renyi}}(m, T)$ (when $m > 1$) . From Eq. (52), we find that $\Sigma^{\text{Renyi}}(m, T)$ reaches its upper bound in the range $T_K < T \leq T_c(m)$. This means that in this range, the lowest free energy state (for a given $T$) completely dominates the contribution to $\Sigma^{\text{Renyi}}(m, T)$.

The above observations also hold for the $p-$spin model, as we derive in section 4.

## 3.5 Special case of the REM

As explained above, the RFEM reduces to the Random Energy Model when $s_0 = 0$ and $\sigma = 0$. In that case we simply have $T_c(m) = m T_K = m J_0 / \sqrt{2 \log 2}$. This can be readily seen from the fact that, when $f_\alpha = \varepsilon_\alpha$ is temperature-independent, the cloned partition function in Eq. (41) satisfies $Z(m, T) = Z(m = 1, T/m)$.

As expected, the complexity becomes equal to the thermodynamic total entropy,

$$\Sigma = \log 2 - \frac{J_0^2}{2T^2} = s_{\text{tot}}, \tag{53}$$

while the vibrational entropy in Eq. (49) vanishes, $s_{\text{vib}} = 0$. The Rényi complexities in Eq. (51) read

$$\Sigma^{\text{Renyi}}(m, T) = \begin{cases} 0 & \text{if} \quad T \leq T_K, \\[2mm] \frac{mJ_0^2}{2(m-1)T^2} \left(1 - \frac{T}{T_K}\right)^2 & \text{if} \quad T_K < T \leq m T_K, \\[2mm] \log 2 - \frac{mJ_0^2}{2T^2} & \text{if} \quad m T_K \leq T \end{cases} \tag{54}$$

and are plotted in Fig 1b.

One may think that the fact that $T_K$ does not depend on $m$ is at odds with the phase diagram drawn by Gardner and Derrida in terms of the number of replicas versus temperature [92]. This difference comes from the fact that they studied the thermodynamics of $\log \overline{Z(m)}$ ($m = \nu$ in their paper) while our Rényi setting uses $\overline{\log Z(m)}$. As will be clear for the $p-$spin model, the former involves a replica symmetric (RS) transition in the regime of $T_K < T \leq mT_K$ when $m > 1$. Yet, RS is not the correct saddle point to compute $\overline{\log Z(m)}$, and we need a 1RSB solution even in the liquid phase. We also note that $\overline{\log Z(m)}$ is related to the scaled cumulant generating function in large deviation theory [77–80], which encodes sample-to-sample fluctuations of (total) free energy density.

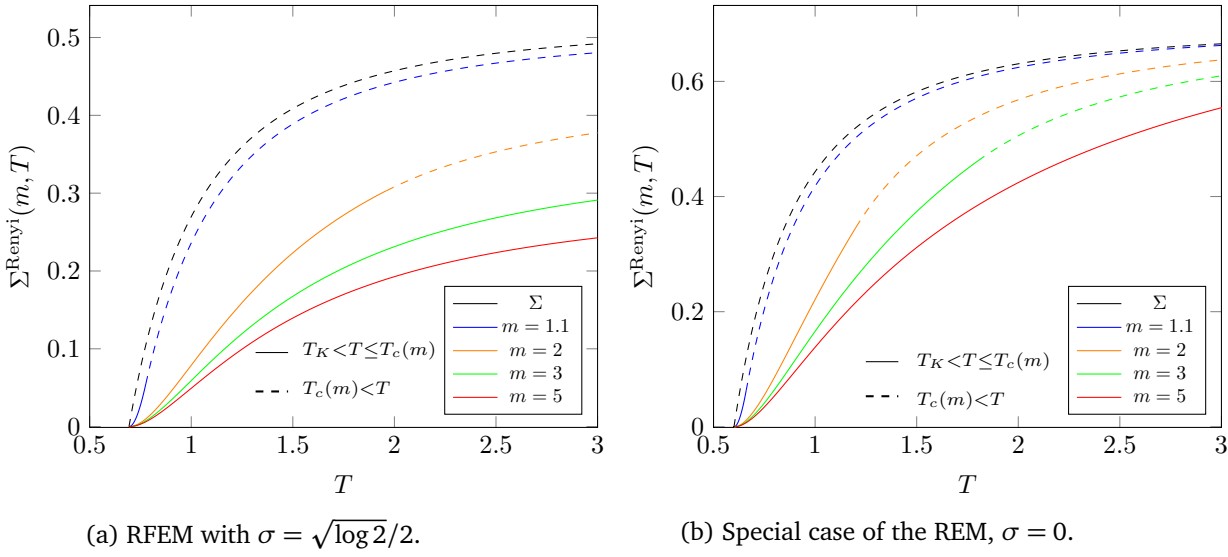

Figure 1: Rényi complexities for the RFEM model in the limit $s_0 \to 0$. The solid (dashed) curves correspond to the regimes below (above) $T_c(m)$ in Eq. (51).

## 4 $p$-spin spherical model

### 4.1 Definition of the model

We next study the $p$-spin spherical model [70], defined by the Hamiltonian,

$$H = - \sum_{1 \leq i_1 < \cdots < i_p \leq N} J_{i_1 \cdots i_p} \sigma_{i_1} \cdots \sigma_{i_p} - h \sum_{i=1}^{N} \sigma_i, \qquad p \geq 3 \tag{55}$$

with $N$ continuous spin variables $\sigma_i \in \mathbb{R}$ that satisfy the spherical constraint $\sum_{i=1}^{N} \sigma_i^2 = N$. The $J_{i_1 \cdots i_p}$ are frozen random couplings drawn from

$$\rho(J_{i_1 \cdots i_p}) = \sqrt{\frac{N^{p-1}}{\pi p!}} \exp\left[ -\frac{1}{2} \frac{2N^{p-1}(J_{i_1 \cdots i_p})^2}{p!} \right]. \tag{56}$$

For introductions to the $p-$spin model see e.g., Refs. [75, 80, 93]. We will focus mainly on the case $p = 3$ with no external field, $h = 0$. In that case the Kauzmann and mode-coupling transition temperatures are given respectively by $T_K \approx 0.586$ and $T_d = \sqrt{3/8} \approx 0.612$.

### 4.2 Replica computation of the Rényi complexity

We recall that the goal is to compute the following Rényi complexity.

$$
\begin{aligned}
\Sigma^{\text{Renyi}}(m) &= \frac{1}{N(1-m)} \overline{\log \sum_\alpha (p_\alpha)^m} = \frac{1}{N(1-m)} \overline{\log \sum_\alpha \left( \frac{e^{-\beta N f_\alpha(T)}}{Z(m=1)} \right)^m} \\
&= \frac{1}{N(1-m)} \left[ \overline{\log Z(m)} - m \overline{\log Z(m=1)} \right] \\
&= \frac{m}{m-1} \left[ \beta \phi(m) - \beta \phi(m=1) \right].
\end{aligned}
\tag{57}
$$

Thus, the main task here is to compute $\beta \phi(m) = -\frac{1}{mN} \overline{\log Z(m)}$, which requires the replica trick:

$$\overline{\log Z(m)} = \lim_{n \to 0} \frac{1}{n} \log \overline{(Z(m))^n}. \tag{58}$$

Using standard techniques (see Ref. [93]), one can express $\overline{(Z(m))^n}$ by

$$\overline{(Z(m))^n} = \left(\prod_{a \neq b} \int dO_{ab}\right) \exp\left[-NG(\{O_{ab}\})\right], \tag{59}$$

$$G(\{O_{ab}\}) = -\frac{\beta^2}{4} \sum_{a,b}^{mn}(O_{ab})^p - \frac{1}{2}\log\det O - \frac{mn}{2}(1 + \log 2\pi), \tag{60}$$

where $G(\{O_{ab}\})$ is the action and $O_{ab}$ is (the elements of) the $mn \times mn$ overlap matrix. The above expressions are obtained by replacing $n$ by $mn$ in the standard computation for the $p$-spin model reviewed in Ref. [93]: the system is now composed of $m$ clones, each having $n$ replicas. We note that the number of replicas per clone, $n$, arises from the replica trick in Eq. (58), with the limit $n \to 0$ applied afterward, whereas the number of clones, $m$, remains a fixed parameter. Thus, clones are sometime referred to as real replicas [71], to distinguish them from the usual replicas counted by $n$. Different values of $m$ will be explored in the phase diagram (see below).

### 4.2.1 Replica ansatz

The overlaps entering the matrix $O$ are then non-trivial, even above $T_K$, and should be carefully determined. Following Refs. [71,94], we assume that $O$ is given by a $m \times m$ block matrix ($m = 3$ in the example below),

$$O = \left[\begin{array}{c|c|c} Q & P & P \\ \hline P & Q & P \\ \hline P & P & Q \end{array}\right], \tag{61}$$

where $Q$ and $P$ are $n \times n$ matrices. $Q$ characterizes overlaps between replicas from the same clone, whereas $P$ represents overlaps between replicas from different clones. Under this assumption, one can express the terms in Eq. (60) by $Q$ and $P$ as follows.

$$\sum_{a,b}^{mn}(O_{ab})^p = m\sum_{a,b}^{n}(Q_{ab})^p + m(m-1)\sum_{a,b}^{n}(P_{ab})^p, \tag{62}$$

$$\begin{aligned}\log\det O &= \log\left[\det\left((Q-P)^{m-1}\right)\det(Q+(m-1)P)\right] \\ &= (m-1)\log\det(Q-P) + \log\det(Q+(m-1)P). \end{aligned} \tag{63}$$

Hence, Eq. (60) becomes

$$\begin{aligned} G(\{Q_{ab}\},\{P_{ab}\}) = &-\frac{\beta^2}{4}\left[m\sum_{a,b}^{n}(Q_{ab})^p + m(m-1)\sum_{a,b}^{n}(P_{ab})^p\right] \\ &-\frac{1}{2}(m-1)\log\det(Q-P) - \frac{1}{2}\log\det(Q+(m-1)P) - \frac{mn}{2}(1+\log 2\pi), \end{aligned} \tag{64}$$

which is valid for generic $Q$ and $P$.

We now make further assumptions in the form of $Q$ and $P$. In particular, we consider the following 1RSB form for $Q$ and $P$ [71,94], composed of submatrices:

$$Q^{1RSB} = \left[\begin{array}{ccc|ccc} 1 & q_1 & q_1 & q_0 & q_0 & q_0 \\ q_1 & 1 & q_1 & q_0 & q_0 & q_0 \\ q_1 & q_1 & 1 & q_0 & q_0 & q_0 \\ \hline q_0 & q_0 & q_0 & 1 & q_1 & q_1 \\ q_0 & q_0 & q_0 & q_1 & 1 & q_1 \\ q_0 & q_0 & q_0 & q_1 & q_1 & 1 \end{array}\right], \quad P^{1RSB} = \left[\begin{array}{ccc|ccc} p_2 & p_1 & p_1 & p_0 & p_0 & p_0 \\ p_1 & p_2 & p_1 & p_0 & p_0 & p_0 \\ p_1 & p_1 & p_2 & p_0 & p_0 & p_0 \\ \hline p_0 & p_0 & p_0 & p_2 & p_1 & p_1 \\ p_0 & p_0 & p_0 & p_1 & p_2 & p_1 \\ p_0 & p_0 & p_0 & p_1 & p_1 & p_2 \end{array}\right]. \tag{65}$$

Each submatrix has size $x \times x$ ($n = 6$ and $x = 3$ in the above example). We assume that $Q^{1\mathrm{RSB}}$ and $P^{1\mathrm{RSB}}$ share the same value of parameter $x$.

For $Q^{1\mathrm{RSB}}$, the diagonal elements correspond to the overlap between the same replicas from the same clone, which is set to one. Off-diagonal elements instead correspond to overlaps between different replicas from the same clone, parameterized by $q_0$ and $q_1$. For $P^{1\mathrm{RSB}}$, instead, the diagonal elements $p_2$ correspond to the overlap between the same replica index yet from different clones. Off-diagonal elements, $p_0$ and $p_1$, are overlaps between different replicas from different clones. Note that when computing the partition function by the saddle-point approximation (see next subsection), the overlaps, as well as the submatrix size $x$, become variational parameters, whereas the number of clones $m$ remains fixed.

We next compute the terms in Eq. (64) using the 1RSB matrices $Q^{1\mathrm{RSB}}$ and $P^{1\mathrm{RSB}}$. One finds

$$\sum_{a,b}^{n} \left(Q_{ab}^{1\mathrm{RSB}}\right)^p = n + n(x-1)(q_1)^p + n(n-x)(q_0)^p, \tag{66}$$

$$\sum_{a,b}^{n} \left(P_{ab}^{1\mathrm{RSB}}\right)^p = n(p_2)^p + n(x-1)(p_1)^p + n(n-x)(p_0)^p, \tag{67}$$

and

$$\log\det\left(Q^{1\mathrm{RSB}} - P^{1\mathrm{RSB}}\right) = d_1 \log\Lambda_1 + d_2 \log\Lambda_2 + d_3 \log\Lambda_3 + n\log(1-p_2), \tag{68}$$

where $\Lambda_1$, $\Lambda_2$, and $\Lambda_3$ are eigenvalues of $\left(Q^{1\mathrm{RSB}} - P^{1\mathrm{RSB}}\right)/(1-p_2)$, given by

$$\Lambda_1 = \frac{1-(q_1-p_1)-p_2}{1-p_2}, \tag{69}$$

$$\Lambda_2 = \Lambda_1 + x\frac{(q_1-p_1)-(q_0-p_0)}{1-p_2}, \tag{70}$$

$$\Lambda_3 = \Lambda_2 + n\frac{q_0-p_0}{1-p_2}, \tag{71}$$

$$\tag{72}$$

with degeneracies $d_1 = n(1-x^{-1})$, $d_2 = n/x - 1$, and $d_3 = 1$. Similarly, one obtains

$$\log\det\left(Q^{1\mathrm{RSB}} + (m-1)P^{1\mathrm{RSB}}\right) = d_1 \log\Lambda_1' + d_2 \log\Lambda_2' + d_3 \log\Lambda_3' + n\log\left(1+(m-1)p_2\right), \tag{73}$$

where $\Lambda_1'$, $\Lambda_2'$, and $\Lambda_3'$ are the eigenvalues of $\left(Q^{1\mathrm{RSB}} + (m-1)P^{1\mathrm{RSB}}\right)/(1+(m-1)p_2)$, given by

$$\Lambda_1' = \frac{1-q_1-(m-1)p_1+(m-1)p_2}{1+(m-1)p_2}, \tag{74}$$

$$\Lambda_2' = \Lambda_1' + x\frac{q_1+(m-1)p_1-q_0-(m-1)p_0}{1+(m-1)p_2}, \tag{75}$$

$$\Lambda_3' = \Lambda_2' + n\frac{q_0+(m-1)p_0}{1+(m-1)p_2}. \tag{76}$$

$$\tag{77}$$

The case of zero external field, that we consider here, corresponds to setting $q_0 = 0$ and $p_0 = 0$. In this case, the $n$ dependence in the action factorizes as
$G\left(\{Q_{ab}^{1\mathrm{RSB}}\}, \{P_{ab}^{1\mathrm{RSB}}\}\right) = G(n, m, x, q_1, p_1, p_2) = n\,g(m, x, q_1, p_1, p_2)$. Here, $g(m, x, q_1, p_1, p_2)$ is

given by

$$
\begin{aligned}
g(m, x, q_1, p_1, p_2) \;=\; & -\frac{m\beta^2}{4}\left[1+(x-1)(q_1)^p+(m-1)(p_2)^p+(m-1)(x-1)(p_1)^p\right] \\
& -\frac{(m-1)}{2}\left[(1-x^{-1})\log\left(1-(q_1-p_1)-p_2\right)+x^{-1}\log\eta_0\right] \\
& -\frac{1}{2}\left[(1-x^{-1})\log\eta_1+x^{-1}\log\eta_2\right]-\frac{m}{2}(1+\log 2\pi),
\end{aligned}
\tag{78}
$$

where we introduce

$$
\begin{aligned}
\eta_0 &= 1+(x-1)(q_1-p_1)-p_2, & (79) \\
\eta_1 &= 1-q_1-(m-1)p_1+(m-1)p_2, & (80) \\
\eta_2 &= 1+(x-1)q_1+(m-1)(x-1)p_1+(m-1)p_2. & (81)
\end{aligned}
$$

### 4.2.2 Saddle-point solutions

Having prepared all detailed equations for the $p$-spin model, we now perform the saddle-point evaluation for $\overline{(Z(m))^n}$ when $N\gg 1$ under the above 1RSB ansatz:

$$
\overline{(Z(m))^n}\approx\exp\left[-N\operatorname*{extr}_{x,q_1,p_1,p_2}\{G(n,m,x,q_1,p_1,p_2)\}\right]=\exp\left[-nN\operatorname*{extr}_{x,q_1,p_1,p_2}\{g(m,x,q_1,p_1,p_2)\}\right].
\tag{82}
$$

Consequently, we obtain the free energy per spin, $\beta\phi(m)$, as

$$
\beta\phi(m)=-\frac{1}{mN}\overline{\log Z(m)}=-\frac{1}{mN}\lim_{n\to 0}\frac{1}{n}\log\overline{(Z(m))^n}=m^{-1}\operatorname*{extr}_{x,q_1,p_1,p_2}\{g(m,x,q_1,p_1,p_2)\}.
\tag{83}
$$

The second term in the Rényi entropy in Eq. (57) is then

$$
\beta\phi(m=1)=\operatorname*{extr}_{x,q_1}\{g(m=1,x,q_1)\},
\tag{84}
$$

$$
\begin{aligned}
g(m=1,x,q_1)=-\frac{\beta^2}{4}\left[1+(x-1)(q_1)^p\right]-\frac{1}{2}\Big[(1-x^{-1})\log(1-q_1) \\
+x^{-1}\log\left(1+(x-1)q_1\right)\Big]-\frac{1}{2}(1+\log 2\pi).
\end{aligned}
\tag{85}
$$

When $T_K\le T$ the solution is

$$
\beta\phi(m=1)=g(m=1,x_*=1,q_{1*}=0)=-\frac{\beta^2}{4}-\frac{1}{2}(1+\log 2\pi).
\tag{86}
$$

Eventually, we are interested in computing the Rényi complexity for fixed $m$, as a function of the clone overlap $p_2$, since this corresponds to the free energy difference as shown in Eq. (26), namely,

$$
\Sigma^{\text{Renyi}}(m,p_2)=\frac{m}{m-1}\left[\beta\phi(m,p_2)-\beta\phi(m=1)\right],
\tag{87}
$$

where $\beta\phi(m,p_2)$ is given by

$$
\beta\phi(m,p_2)=m^{-1}\operatorname*{extr}_{x,q_1,p_1}\{g(m,x,q_1,p_1,p_2)\}.
\tag{88}
$$

Thus computing $\Sigma^{\text{Renyi}}(m,p_2)$ boils down to finding $x_*$, $q_{1*}$, and $p_{1*}$ which extremizes the function $g(m,x,q_1,p_1,p_2)$, given $m$ and $p_2$. We solved the coupled saddle-point equations numerically and analytically, which leads to the following three distinct regimes, depending on the value of $m$ and $p_2$.

- RS regime: $x_* = 1$ (equivalently, $q_{1*} = p_{1*} = 0$).

- $\text{RSB}_a$ regime: $x_* < 1$ and $q_{1*} > p_{1*} > 0$.

- $\text{RSB}_b$ regime: $x_* < 1$ and $q_{1*} = p_{1*} > 0$.

The phase diagram in the $m$ versus $p_2$ plane for three values of $T$ is shown in Fig. 2. The boundary between RS and either $\text{RSB}_a$ or $\text{RSB}_b$ is denoted as $p_2^{(1)}(m)$, whereas the boundary between $\text{RSB}_a$ and $\text{RSB}_b$ is denoted as $p_2^{(2)}(m)$.

To better understand the phase diagram, we now monitor the solutions, $x_*$, $q_{1*}$, and $p_{1*}$, along representative paths in the phase diagram, $m = 2.5$ at $T = 0.6 < T_d$ in Fig. 3a and at $T = 0.75 > T_d$ in Fig. 3b. For $T = 0.6$, at low values of $p_2$ the trivial solution is $x_* = 1$ (equivalently, $q_{1*} = p_{1*} = 0$), which corresponds to the replica symmetric (RS) ansatz (RS regime). At intermediate values of $p_2$ with $p_2^{(1)} < p_2 < p_2^{(2)}$, a non-trivial, one-step replica symmetry broken solution appears with $x_* < 1$ and $q_{1*} > p_{1*} > 0$ ($\text{RSB}_a$ regime). At high values of $p_2 > p_2^{(2)}$, the solutions, $q_{1*}$ and $p_{1*}$, merge, while $x_* < 1$ ($\text{RSB}_b$ regime). As we present in details in Appendix A (for the case of $p = 3$), both the RS and $\text{RSB}_b$ solutions can be found analytically. In the intermediate $\text{RSB}_a$ regime instead, we resorted to numerical extremization.

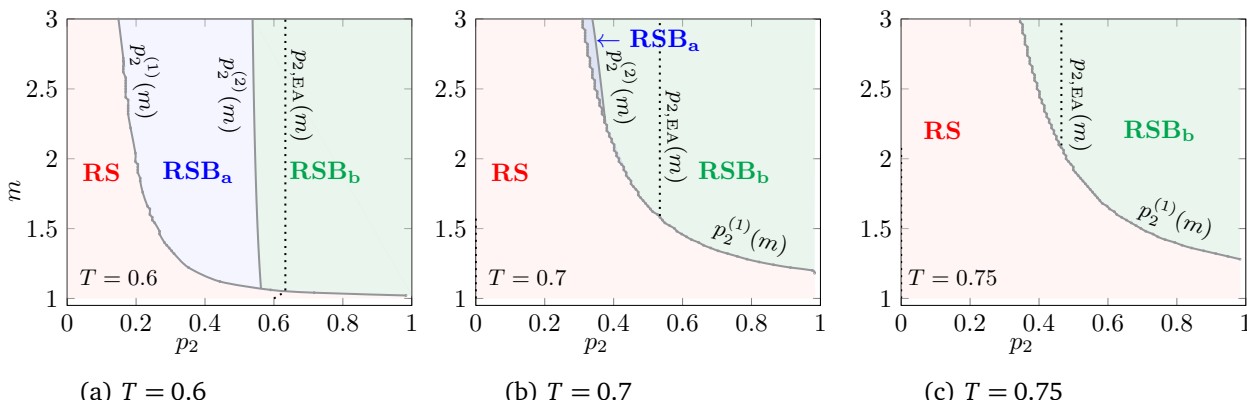

|  |  |  |
|---|---|---|
| (a) $T = 0.6$ | (b) $T = 0.7$ | (c) $T = 0.75$ |

Figure 2: Phase diagrams of the spherical $p = 3$ spin model for three different temperatures, $T_K < T = 0.6 < T_d$ (a), $T_d < T = 0.7$ (b), and $T_d < T = 0.75$ (c). $p_{2,\text{EA}}$ locates the local minimum of $\Sigma^{\text{Renyi}}(m, p_2)$, and lies either in the RS or in the $\text{RSB}_b$ regimes. Note that the line $m = 1$ always lies in the RS regime, so that the (Shannon) complexity $\Sigma$ is given at all $T$ by the RS solution.

At high enough temperature, e.g., at $T = 0.7 > T_d$ shown in Fig. 3b, the intermediate $\text{RSB}_a$ regime disappears, as is also visible in Fig 2c. Hence $x_* < 1$ and $q_{1*} = p_{1*} > 0$ above $p_2^{(1)}$.

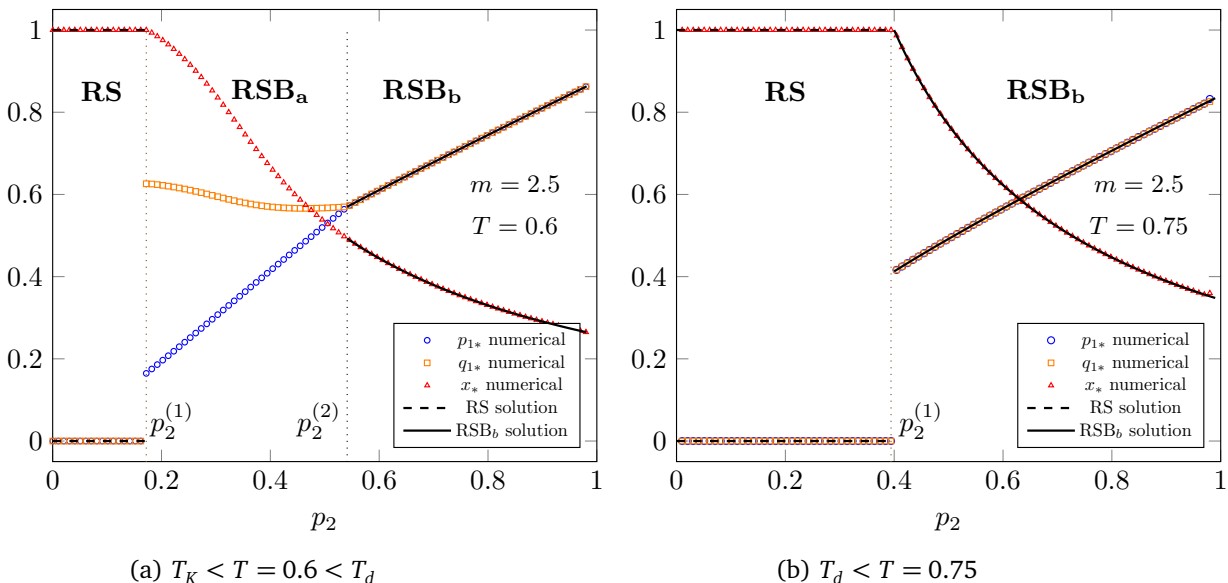

(a) $T_K < T = 0.6 < T_d$               (b) $T_d < T = 0.75$

Figure 3: Saddle points, $x_*$ (triangles), $q_{1*}$ (squares), and $p_{1*}$ (diamond), as functions of $p_2$ for a given $m = 2.5$, below (a) and above (b) $T_d$. The dashed and solid curves correspond to the analytical solutions obtained in the RS and RSB$_b$ regimes, respectively.

Once the saddle points have been identified, we obtain $\Sigma^{\text{Renyi}}(m, p_2)$ as a function of $p_2$. Figure 4 shows $\Sigma^{\text{Renyi}}(m, p_2)$ for several values of $m$ and $T$, where the circles are the numerical solutions, while the dashed and solids curves correspond to the analytic solutions from the RS and RSB$_b$ regimes, respectively. The analytic solutions reproduce correctly numerical solutions at the range of low (RS) and high (RSB$_b$) $p_2$ values, while they do not capture the intermediate values of $p_2$ (RSB$_a$). This is especially visible at higher $m$ and lower temperatures, reflecting the phase diagram in Fig. 2. Importantly, we find that the value $p_{2,\text{EA}}(T)$ that locates the local minimum of $\Sigma^{\text{Renyi}}(m, p_2)$ lies in the RS or RSB$_b$ region (as shown in Fig. 2 as dashed lines). Therefore, the Rényi complexity, $\Sigma^{\text{Renyi}}(m, T) = \Sigma^{\text{Renyi}}(m, p_{2,\text{EA}}(T))$, can be computed analytically for all $m$.

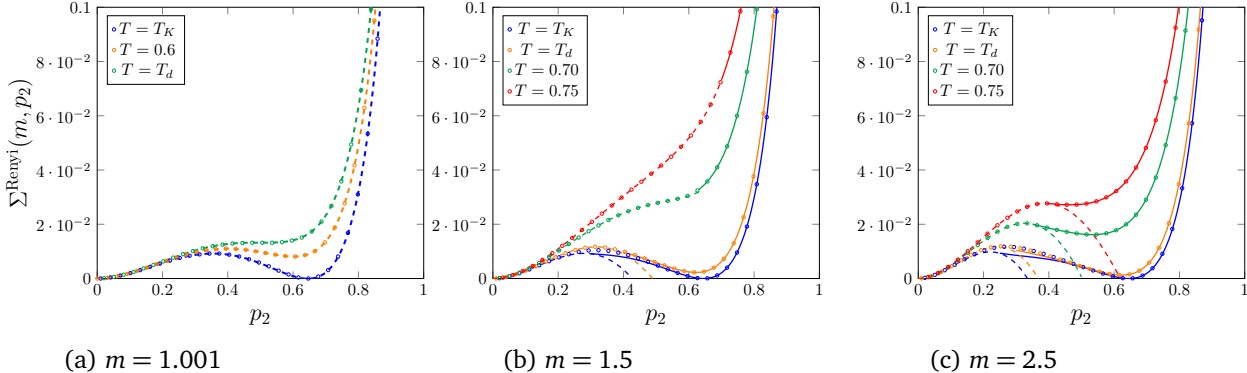

(a) $m = 1.001$          (b) $m = 1.5$          (c) $m = 2.5$

Figure 4: $\Sigma^{\text{Renyi}}(m, p_2)$ with different indices $m$, as functions of $p_2$, for varying temperatures. In each case we show $\Sigma^{\text{Renyi}}(m, p_2)$ computed from numerical minimization (circles), as well as the analytic solutions in the RS (dashed) and RSB$_b$ (solid) regimes. There is an expected discrepancy between the numerical points and the curves in the RSB$_a$ regime where none of the analytic solutions hold. In the $m \to 1$ limit, $\Sigma^{\text{Renyi}}(m, p_2)$ coincides with $\Sigma$ given by Eq. (A.8).

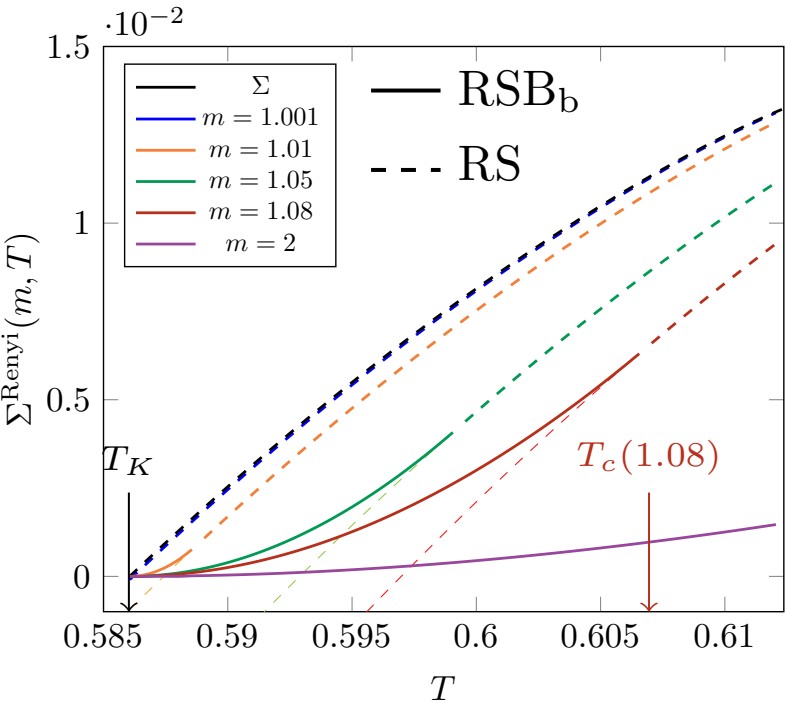

Figure 5: Rényi complexity for different indices $m$, as functions of $T$. Above $T_c(m)$, $\Sigma^{\text{Renyi}}$ is computed by using the RS solution in Eq. (A.7) (dashed-curves), which vanishes above $T_K$. Below $T_c(m)$, the $\text{RSB}_b$ solution using Eq. (A.28) is needed to compute $\Sigma^{\text{Renyi}}$ (solid-curves). We also plot the (Shannon) complexity $\Sigma$ as the black dashed curve. The locations of $T_K = T_c(m \to 1)$ and $T_c(m = 1.08)$ are indicated with vertical arrows.

Finally, we plot $\Sigma^{\text{Renyi}}(m, T) = \Sigma^{\text{Renyi}}(m, p_{2,\text{EA}}(T))$ in Fig. 5, where $p_{2,\text{EA}}(T)$ is determined in the RS and $\text{RBS}_b$ regimes. We find that $\Sigma^{\text{Renyi}}(m, T)$, calculated using the RS solutions (dashed curves), decreases as $T$ decreases in a concave way and becomes zero above $T_K$ for $m > 1$. To compute the Rényi complexity correctly, the $\text{RSB}_b$ solution (solid curves) must be used. This solution appears below an $m$-dependent temperature, $T_c(m)$, defined by Eq. (A.34) in Appendix A. With the $\text{RSB}_b$ solution, $\Sigma^{\text{Renyi}}(m, T)$ decreases in a convex way at lower temperatures and becomes zero at the same temperature, $T_K = T_c(m = 1)$, regardless of the value of $m$. This behavior is also observed in the RFEM, as discussed in Sec. 3. In Fig 6 we show larger values of $m$, on the full temperature range $T_K < T < T_{\text{max}}$, where $T_{\text{max}}$ is the maximum temperature at which the local minimum and hence $p_{2,\text{EA}}$ exist. (cf. A.2). Besides, we show in Sec. A.2 of Appendix A that, as was made explicit for the RFEM (see Eq. (52)), the Rényi complexity below $T_c(m)$ is essentially given by $\Sigma^{\text{Renyi}}_{\infty}(T)$ (min-entropy), namely,

$$\Sigma^{\text{Renyi}}(m, T) = \frac{m}{m-1} \Sigma^{\text{Renyi}}_{\infty}(T) \qquad \left( T_K < T < T_c(m), \, m > 1 \right). \qquad (89)$$

Therefore, as discussed in the RFEM case, the lowest free energy state (at a given $T$) entirely dominates the contribution to $\Sigma^{\text{Renyi}}(m, T)$.

We can rewrite Eq. (89) in two interesting ways. First we can express all Rényi complexities for $m > 1$ in terms of the Rényi complexity with $m = 2$ over a restricted temperature range,

$$\Sigma^{\text{Renyi}}(m, T) = \frac{m}{2(m-1)} \Sigma^{\text{Renyi}}(m = 2, T), \qquad T_K < T \leq \min\{T_c(m), T_c(m = 2)\}. \qquad (90)$$

As mentioned before, the Rényi complexity with index $m = 2$ corresponds to the annealed Franz-Parisi potential (see Eq. (27)) which is the easiest to compute in numerical simulations.

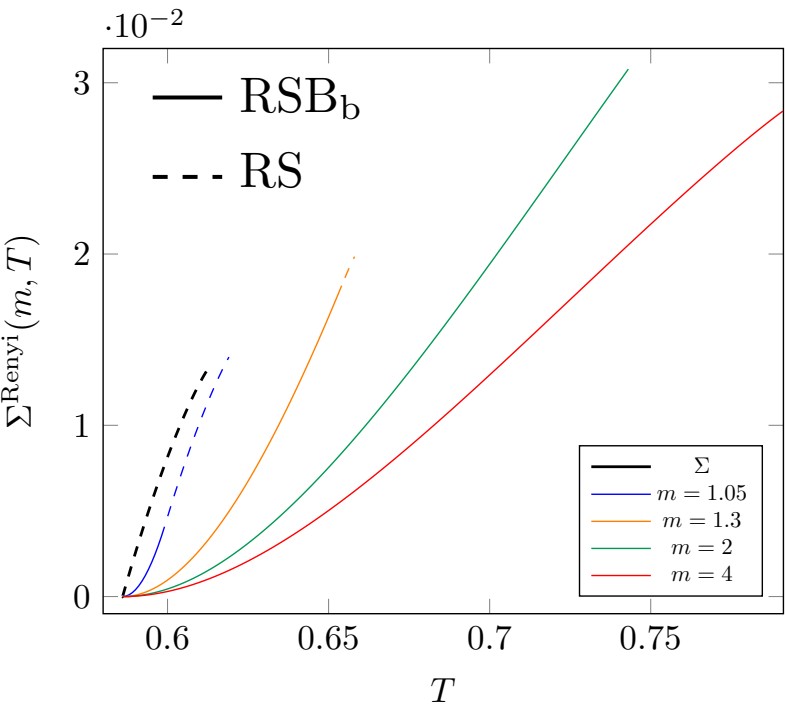

Figure 6: Rényi complexity with large indices for $T_K < T < T_{\max}$.

Moreover, Eq. (89) implies that the ratio of Rényi complexities with different indices, $m_1$ and $m_2$, with $1 < m_1 < m_2$ is a constant below $T_c(m_1)$,

$$\Sigma^{\text{Renyi}}(m_1, T)/\Sigma^{\text{Renyi}}(m_2, T) = \frac{m_1(m_2 - 1)}{m_2(m_1 - 1)}, \qquad T_K < T \leq T_c(m_1). \tag{91}$$

In numerical experiments, this could be used to detect the transition point, $T_c(m)$, from the RS to RSB$_b$ regimes, provided that Eq. (89) would hold beyond the mean-field limit.

## 5 Conclusion and Discussion

We have computed the Rényi entropy version of complexity, Rényi complexity, for prototypical mean-field disordered models: the random energy model, the random free energy model, and the $p$-spin spherical model. We first demonstrated that the Rényi complexity with the Rényi index $m$ is linked to the free energy difference of the generalized $m$-component annealed Franz-Parisi potential. Detailed calculations of Rényi complexity for the random energy model and random free energy model were performed without using the replica trick, yet these computations suggest that replica symmetry-breaking solutions are required even in the liquid phase. We then performed replica computations for the $p$-spin spherical model using techniques involving $m$ clones (real replicas) and $n$ replicas. We confirmed that indeed replica symmetry-breaking solutions are needed in the liquid phase when $m > 1$. All models studied consistently exhibit that all Rényi complexities with $m > 1$ vanish at the same Kauzmann transition temperature $T_K$, separating the liquid and glass phases, irrespective of the value of $m$. This finding suggests that the Rényi complexity is also a useful observable for estimating or locating $T_K$ in practical applications when measured in the liquid phase and extrapolated toward lower temperatures.

For practical measurements of Rényi complexity, through Eq. (27), one can compute the $m$-component annealed Franz-Parisi potential, which can be achieved by a generalization of what

has been done numerically, e.g., for glass-forming liquids [36–38]. However, our mean-field computations in this paper suggest that sampling becomes more challenging when $T$ is low and $m$ is large, due to the underlying putative replica symmetry breaking (RSB) at $T_c(m)$, at least at the mean-field level. It would be interesting to investigate whether the features observed in our mean-field study persist in finite-dimensional systems. Kurchan and Levine proposed a different way to measure the Rényi complexity by enumerating frequently appearing local patches in amorphous configurations. In principle, this method would not be affected by the sampling problem (in terms of measuring Rényi complexity), and is insightful as it connects a real-space perspective (an inherently finite-dimensional property) with Rényi complexity. It could also allow to verify whether the relation between the Rényi complexities with arbitrary index $m$ and the annealed Franz-Parisi potential shown in Eq. (90), as well as Eq. (91), hold in finite-dimensional systems.

In this paper, we considered mainly the case $m > 1$, motivated by the practical use of measurement of the Rényi entropy with, say, $m = 2, 3, \cdots$. In general, varying the Rényi index $m$ from the Shannon limit $m \to 1$ corresponds to biasing ($m > 1$) or unbiasing ($m < 1$) the original probability distribution. Thus, similar to the large deviation studies, it is interesting to extend our computation to $0 < m < 1$ (or even negative $m$). It would also be interesting to compute the Rényi complexity for more complicated mean-field models, such as the mixed $p$-spin model [75, 95, 96], and replica liquid theory [97], where the complexity plays a crucial role in understanding the glassy behavior of the system.

## Acknowledgements

We thank Ludovic Berthier, Silvio Franz, Jorge Kurchan, Mauro Pastore, and Gilles Tarjus for interesting discussions. We are also grateful to Silvio Franz for sharing his extremization code with us.

**Funding information** NJ and MO have been supported by a grant from MIAI@Grenoble Alpes (ANR-19-P3IA-0003).

## A Analytical solution of the $p-$spin model

In this appendix, we describe the detailed calculations leading to the determination of the saddle-point solutions. In particular, we give the analytical solution for $\Sigma^{\text{Renyi}}(m, T)$, in the case of $p = 3$.

We wish to find the saddle-point solution for $g(m, x, q_1, p_1, p_2)$ in Eq. (78), given $m$ and $p_2$. The derivatives of $g(m, x, q_1, p_1, p_2)$ with respect to $x$, $q_1$, $p_1$, and $p_2$ are respectively given

by

$$\frac{\partial g(m,x,q_1,p_1,p_2)}{\partial x} = -\frac{m\mu}{2p}\left[(q_1)^p + (m-1)(p_1)^p\right] - \frac{(m-1)}{2x^2}\log\left(\frac{1-(q_1-p_1)-p_2}{\eta_0}\right)$$
$$-\frac{(m-1)(q_1-p_1)}{2x\eta_0} - \frac{1}{2x^2}\log(\eta_1/\eta_2) - \frac{q_1+(m-1)p_1}{2x\eta_2}, \tag{A.1}$$

$$\frac{\partial g(m,x,q_1,p_1,p_2)}{\partial q_1} = \frac{(1-x)}{2}\left[m\mu(q_1)^{p-1} - \frac{(m-1)(q_1-p_1)}{\eta_0(1-(q_1-p_1)-p_2)} - \frac{q_1+(m-1)p_1}{\eta_1\eta_2}\right], \tag{A.2}$$

$$\frac{\partial g(m,x,q_1,p_1,p_2)}{\partial p_1} = \frac{(1-x)(m-1)}{2}\left[m\mu(p_1)^{p-1} - \frac{q_1+(m-1)p_1}{\eta_1\eta_2} + \frac{q_1-p_1}{\eta_0(1-(q_1-p_1)-p_2)}\right], \tag{A.3}$$

$$\frac{\partial g(m,x,q_1,p_1,p_2)}{\partial p_2} = \frac{m-1}{2}\left[-m\mu(p_2)^{p-1} + \frac{1}{x\eta_0} + \frac{1-x}{x}\frac{1}{\eta_1} - \frac{1}{x\eta_2} + \frac{x-1}{x}\frac{1}{1-(q_1-p_1)-p_2}\right], \tag{A.4}$$

where $\mu = \beta^2 p/2$ and

$$\eta_0 = 1 - p_2 + (x-1)(q_1-p_1), \tag{A.5a}$$
$$\eta_1 = 1 + (m-1)p_2 - (m-1)p_1 - q_1, \tag{A.5b}$$
$$\eta_2 = 1 + (m-1)p_2 + (m-1)(x-1)p_1 + (x-1)q_1. \tag{A.5c}$$

## A.1   RS solution

One can easily check that the saddle-point conditions given by Eqs. (A.1), (A.2), and (A.3) have the trivial solution, $x_* = 1$ (or $q_{1*} = 0$ and $p_{1*} = 0$), which corresponds to the replica symmetric ansatz. Hence, for $m > 0$ with $m \neq 1$, the last variational equation in Eq. (A.4) becomes

$$\mu(p_2)^{p-1} = \frac{p_2}{(1-p_2)[1+(m-1)p_2]}. \tag{A.6}$$

$p_{2*} = 0$ is the trivial solution of Eq. (A.6), which corresponds to the liquid state. Yet we wish to find a non-trivial solution in the local minimum of $g(m,x_* = 1, q_{1*} = 0, p_{1*} = 0, p_2)$, which corresponds to the Edwards-Anderson parameter, $p_{2,\text{EA}} > 0$, characterizing the metastable glass state. For $p = 3$ the RS solution then reads

$$x_*^{\text{RS}} = 1, \tag{A.7a}$$
$$q_{1*}^{\text{RS}} = p_{1*}^{\text{RS}} = 0, \tag{A.7b}$$
$$p_{2,\text{EA}}^{\text{RS}}(m,T) = R_1\left[-\frac{3}{2}\beta^2(m-1), \frac{3}{2}\beta^2(m-2), \frac{3}{2}\beta^2, -1\right], \tag{A.7c}$$

where $R_1$ is the real root in Eq. (A.36) of the order-3 polynomial in Eq. (A.6).

When $m \to 1$, we obtain the complexity as

$$\Sigma(p_{2,\text{EA}}^{\text{RS}}) = \lim_{m\to 1}\frac{m}{m-1}\left[\beta\phi(m, p_{2,\text{EA}}^{\text{RS}}) - \beta\phi(m=1)\right]$$
$$= -\frac{\beta^2}{4}(p_{2,\text{EA}}^{\text{RS}})^p - \frac{1}{2}\log(1-p_{2,\text{EA}}^{\text{RS}}) - \frac{p_{2,\text{EA}}^{\text{RS}}}{2}, \tag{A.8}$$

where we used Eq. (86) and L'Hopital's rule to evaluate the limit. In that case Eq. (A.6) becomes an order-2 polynomial and

$$p_{2,\text{EA}}^{\text{RS}}(T) = \frac{1}{2} + \frac{1}{2}\sqrt{1 - \frac{8T^2}{3}}.$$

(A.9)

## A.2  RSB$_b$ solution

Finding non-trivial solutions, namely, $x_* < 1$, $q_{1*} > 0$, and $p_{1*} > 0$ requires solving the coupled saddle-point equations given by Eqs. (A.1-A.4). When $m \neq 1$, they become

$$0 = \frac{m\mu}{p}[(q_1)^p + (m-1)(p_1)^p] + \frac{(m-1)}{x^2}\log\left(\frac{1-(q_1-p_1)-p_2}{\eta_0}\right)$$
$$+ \frac{(m-1)(q_1-p_1)}{x\eta_0} + x^{-2}\log(\eta_1/\eta_2) + \frac{q_1+(m-1)p_1}{x\eta_2},$$

(A.10)

$$0 = m\mu(q_1)^{p-1} - \frac{(m-1)(q_1-p_1)}{\eta_0(1-(q_1-p_1)-p_2)} - \frac{q_1+(m-1)p_1}{\eta_1\eta_2},$$

(A.11)

$$0 = m\mu(p_1)^{p-1} - \frac{q_1+(m-1)p_1}{\eta_1\eta_2} + \frac{q_1-p_1}{\eta_0(1-(q_1-p_1)-p_2)},$$

(A.12)

$$0 = -m\mu p_2^{p-1} + \frac{1}{x}\frac{1}{\eta_0} + \frac{1-x}{x}\frac{1}{\eta_1} - \frac{1}{x}\frac{1}{\eta_2} + \frac{x-1}{x}\frac{1}{1-(q_1-p_1)-p_2}.$$

(A.13)

While a fully general analytical solution to the above equations is out of reach, they can be solved in the RSB$_b$ regime, where $q_{1*} = p_{1*} > 0$. This allows us to compute analytically the Rényi complexities $\Sigma^{\text{Renyi}}(m, T)$, as the location of the local minimum, $p_{2,\text{EA}}$, of $\Sigma^{\text{Renyi}}(m, p_2)$ is always located in the RS or RSB$_b$ regimes (cf. Fig. 2 and Fig. 4).

When $q_1 = p_1$ and for $m > 0$ with $m \neq 1$, Eqs. (A.10-A.13) reduce to

$$\frac{\mu}{p}(q_1)^p = -\frac{1}{x m}\frac{q_1}{\eta_2} - \frac{1}{m^2 x^2}\log(\eta_1/\eta_2),$$

(A.14)

$$\mu(q_1)^{p-1} = \frac{q_1}{\eta_1\eta_2},$$

(A.15)

$$\mu(p_2)^{p-1} = \frac{q_1}{\eta_1\eta_2} + \frac{1}{m}\frac{\eta_1-\eta_0}{\eta_0\eta_1},$$

(A.16)

and Eqs. (A.5) to

$$\eta_1 = 1 - mq_1 + (m-1)p_2,$$

(A.17)

$$\eta_2 = 1 + m(x-1)q_1 + (m-1)p_2.$$

(A.18)

Equations (A.14) and (A.15) can be rewritten as

$$\frac{(1-y)^2}{py} + \log y + 1 - y = 0,$$

(A.19)

$$\mu(q_1)^{p-2}(\eta_1)^2 - y = 0,$$

(A.20)

where $y = \eta_1/\eta_2$. For a given $p$, one can obtain $y$ by solving Eq. (A.19) via, e.g., the bisection method. For $p = 3$, $y \approx 0.3549927$. Then Eq. (A.20) gives rise to the solution, $q_{1*}(m, T, p_2)$. Finally, we obtain $x_*$ by inverting the relation, $y = \eta_1/\eta_2$, and find

$$x_*(m, T, p_2) = \frac{(1-y)(1-mq_{1*}+(m-1)p_2)}{myq_{1*}}.$$

(A.21)

We next find the Edwards-Anderson parameter $p_{2,\text{EA}}$, locating the local minimum associated with the metastable glass state. Subtracting Eq. (A.15) from Eq. (A.16) gives

$$\mu\big[(p_2)^{p-1}-(q_1)^{p-1}\big]=\frac{p_2-q_1}{\eta_0\,\eta_1}. \tag{A.22}$$

We now specialize to the case of $p=3$, where the above equation becomes

$$\mu(p_2-q_1)(p_2+q_1)=\frac{p_2-q_1}{\eta_0\,\eta_1}. \tag{A.23}$$

One solution is $p_2=q_1$. We argue that this is the only correct solution (using proof by contradiction). Indeed if $p_2\neq q_1$ we have

$$\mu(p_2+q_1)=\frac{1}{\eta_0\,\eta_1}\Leftrightarrow\mu(p_2+q_1)(1-p_2)(1+(m-1)p_2-m\,q_1)-1=0. \tag{A.24}$$

This is a second order polynomial for $q_1$. However one can check that the discriminant,

$$\Delta=\mu^2(p_2-1)^4+4m\,\mu(p_2-1)\big[1+\mu\,p_2(p_2-1)(1+(m-1)p_2)\big], \tag{A.25}$$

is negative for all $p_2$ (for arbitrary values of $\beta$, $m$), so that there cannot exist any real solution for $q_1$ if $p_2\neq q_1$. Therefore $p_2=q_1$.

Assuming then that $p_2=q_1$, we can rewrite Eq. (A.20) as

$$\mu\,p_2(1-p_2)^2-y=0. \tag{A.26}$$

The order-3 polynomial has the solution $p_{2,\text{EA}}(T)=R_2[\mu,-2\mu,\mu,-y]$, which is independent of $m$.

We then summarize the $\text{RSB}_b$ solution for $p=3$, by expressing $x_*^{\text{RSB}_b}$, $q_{1*}^{\text{RSB}_b}$, and $p_{1*}^{\text{RSB}_b}$ as a function of $p_2$,

$$q_{1*}^{\text{RSB}_b}(m,\beta,p_2)=p_{1*}^{\text{RSB}_b}(m,\beta,p_2)$$
$$=R_2\left[\frac{3}{2}\beta^2m^2,-3m\beta^2(1+[m-1]p_2),\frac{3}{2}\beta^2(1+[m-1]p_2)^2,-y\right], \tag{A.27a}$$

$$x_*^{\text{RSB}_b}(m,\beta,p_2)=\frac{1-y}{y}\frac{1+(m-1)p_2-m\,q_1^{\text{RSB}_b}(m,\beta,p_2)}{m\,q_1^{\text{RSB}_b}(m,\beta,p_2)}, \tag{A.27b}$$

and as a function of $\beta=1/T$,

$$q_{1*}^{\text{RSB}_b}(\beta)=p_{1*}^{\text{RSB}_b}(\beta)=p_{2,\text{EA}}^{\text{RSB}_b}(\beta), \tag{A.28a}$$

$$p_{2,\text{EA}}^{\text{RSB}_b}(\beta)=R_2\left[\frac{3}{2}\beta^2,-3\beta^2,\frac{3}{2}\beta^2,-y\right], \tag{A.28b}$$

$$x_*^{\text{RSB}_b}(m,\beta)=\frac{1-y}{y}\frac{1-p_{2,\text{EA}}^{\text{RSB}_b}(\beta)}{m\,p_{2,\text{EA}}^{\text{RSB}_b}(\beta)}. \tag{A.28c}$$

Note that $p_{2,\text{EA}}^{\text{RSB}_b}$ is defined up to a certain temperature $T_{\max}$, such that the polynomial root remains real, which can be found for $p=3$ as $T_{\max}=\frac{1}{3}\sqrt{\frac{2}{y}}\approx 0.7912$.

Finally we compute $\Sigma^{\text{Renyi}}(m, p_{2,\text{EA}})$ in the RSB$_b$ regime:

$$
\begin{aligned}
\Sigma^{\text{Renyi}}(m, p_{2,\text{EA}}^{\text{RSB}_b}) &= \frac{m}{m-1}\left[\beta\phi(m, p_{2,\text{EA}}^{\text{RSB}_b}) - \beta\phi(m=1)\right] \\
&= \frac{m}{m-1}\left[-\frac{\beta^2}{4}(mx_*^{\text{RSB}_b}-1)(p_{2,\text{EA}}^{\text{RSB}_b})^3 - \frac{1}{2}\log(1-p_{2,\text{EA}}^{\text{RSB}_b})\right. \\
&\qquad\qquad\left. - \frac{1}{2mx_*^{\text{RSB}_b}}\log\frac{1+(mx_*^{\text{RSB}_b}-1)p_{2,\text{EA}}^{\text{RSB}_b}}{1-p_{2,\text{EA}}^{\text{RSB}_b}}\right].
\end{aligned}
\tag{A.29}
$$

Interestingly, the terms inside the bracket in Eq. (A.29) do not depend on $m$, since from the solutions in Eq. (A.28), $p_{2,\text{EA}}^{\text{RSB}_b}$ and $mx_*^{\text{RSB}_b}$ depend on temperature only. Thus, as we found explicitly for the RFEM, one can express the Rényi complexities below $T_c(m)$ in terms of the min-entropy, $\Sigma_\infty^{\text{Renyi}}(p_{2,\text{EA}}^{\text{RSB}_b})$:

$$
\Sigma^{\text{Renyi}}(m, p_{2,\text{EA}}^{\text{RSB}_b}) = \frac{m}{m-1}\Sigma_\infty^{\text{Renyi}}(p_{2,\text{EA}}^{\text{RSB}_b}),
\tag{A.30}
$$

where $\Sigma_\infty^{\text{Renyi}}(p_{2,\text{EA}}^{\text{RSB}_b})$ is given by

$$
\Sigma_\infty^{\text{Renyi}}(p_{2,\text{EA}}^{\text{RSB}_b}) = -\frac{\beta^2(1-y-p_{2,\text{EA}}^{\text{RSB}_b})(p_{2,\text{EA}}^{\text{RSB}_b})^2}{4y} - \frac{1}{2}\log(1-p_{2,\text{EA}}^{\text{RSB}_b}) + \frac{p_{2,\text{EA}}^{\text{RSB}_b} y\log y}{2(1-p_{2,\text{EA}}^{\text{RSB}_b})(1-y)}.
\tag{A.31}
$$

## A.3  Transition temperature $T_c(m)$

We determine the temperature $T_c(m)$ (below $T_{\max}$) marking the transition between the RS and RSB$_b$ solutions. In the RSB$_b$ regime, as shown in Eq. (A.28), we have $q_{1*}^{\text{RSB}_b} = p_{1*}^{\text{RSB}_b} = p_{2,\text{EA}}^{\text{RSB}_b}$. In this case, the condition for the local minimum, Eq. (A.16), becomes

$$
\mu p_{2,\text{EA}}^{\text{RSB}_b} = \frac{1}{(1-p_{2,\text{EA}}^{\text{RSB}_b})(1+(mx_*^{\text{RSB}_b}-1)p_{2,\text{EA}}^{\text{RSB}_b})}.
\tag{A.32}
$$

By using $y = \eta_1/\eta_2$, we can express $p_{2,\text{EA}}^{\text{RSB}_b}$ in terms of $x_*^{\text{RSB}_b}$ as $p_{2,\text{EA}}^{\text{RSB}_b} = (1-y)/\left[1+y(mx_*^{\text{RSB}_b}-1)\right]$. Therefore, Eq. (A.32) can be rewritten in terms of $x_*^{\text{RSB}_b}$ as

$$
\frac{3}{2T^2} = \frac{[1+y(mx_*^{\text{RSB}_b}-1)]^3}{m^2(x_*^{\text{RSB}_b})^2 y(1-y)}.
\tag{A.33}
$$

From Eq. (A.33), the transition temperature $T_c(m)$ is identified when $x_*^{\text{RSB}_b} \to x_*^{\text{RS}} = 1$. Thus we get

$$
T_c(m) = \sqrt{\frac{3m^2 y(1-y)}{2[1+y(m-1)]^3}}.
\tag{A.34}
$$

In particular, one can check that $T_c(m=1) = T_K$ [80]. In Fig. 7, we plot $T_c(m)$ in the $m$ versus $T$ plane. By solving $T_c(m) = T_{\max} = \frac{1}{3}\sqrt{\frac{2}{y}}$, we find that for $m \geq m_c = \frac{2(1-y)}{y} \approx 3.63$, the Rényi complexity is given by the RSB$_b$ solution on the whole interval, $T_K \leq T \leq T_{\max}$, and there exist no non-trivial ($p_{2,\text{EA}} > 0$) RS regime, as can be seen also in Fig. 6 for $m = 4$.

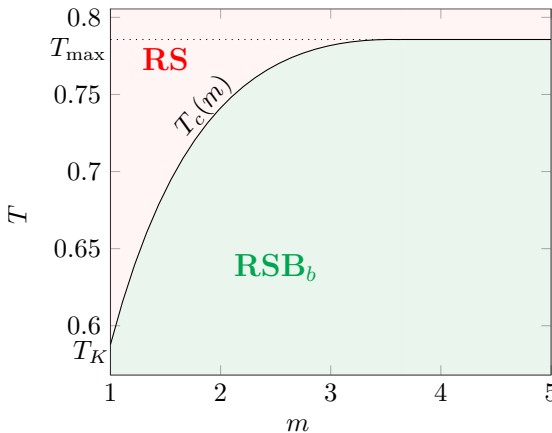

Figure 7: Phase diagram for the computation of the Rényi complexity. $T_c(m)$ separates the RS and RSB$_b$ regimes. Specifically, $T_c(m = 1) = T_K$ and $T_c(m) = T_{max}$ when $m \geq m_c = \frac{2(1-y)}{y} \approx 3.63$.

### A.4 Roots of order 3 polynomials

We write here for reference the solutions of the polynomial equation,

$$a x^3 + b x^2 + c x + d = 0. \tag{A.35}$$

The three roots $x = R_j$ ($j \in \{1, 2, 3\}$) of Eq. (A.35) are given by

$$R_j[a, b, c, d] = P + z_j \left[ Q + \sqrt{Q^2 - (P^2 - R)^3} \right]^{\frac{1}{3}} + \bar{z}_j \left[ Q - \sqrt{Q^2 - (P^2 - R)^3} \right]^{\frac{1}{3}}, \tag{A.36}$$

where

$$P = -\frac{b}{3a} \qquad Q = P^3 + \frac{bc - 3ad}{6a^2} \qquad R = \frac{c}{3a}$$

$$z_1 = 1 \qquad z_2 = -\frac{1}{2}\left(1 + \sqrt{3}i\right) \qquad z_3 = -\frac{1}{2}\left(1 - \sqrt{3}i\right).$$

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
