# Peer review of "Rényi complexity in mean-field disordered systems"

_SciPost Physics_

## Round 1 · Referee Report · Anonymous (Referee 1) · 2025-2-2

Strengths

1- The paper provides a novel mapping between an information-theoretical quantity, the Rényi entropy, and a statistical mechanics concept, the annealed Franz-Parisi potential, providing thus a common ground for further advancements in the interpretation and formalization of disordered systems.

2- In view of the aforementioned connection between two different fields, the authors give a clear and complete overview of both, stressing the reasons for considering such quantities that generalize the concept of entropy in information theory and physics.

3- The analysis of the models they consider as running examples for their approach is complete, well explained and consistent with previous results.

4- The observation on the difference between their calculation and the one to obtain the large deviations of the free energy is interesting for the community working on that topic.

Weaknesses

1- Minor weaknesses in the presentation (see below, Requested changes)

Report

The paper under review considers the concept of Rényi complexity in the context of disordered systems. After a review on how complexity (configurational entropy) in spin glasses has been theoretically evaluated in the past, and on the generalized definition of complexity represented by the Rényi entropy, the authors manage to identify the Rényi complexity with index $m$ with a rescaled annealed Franz-Parisi potential for $m$-clones, a powerful tool used in the past to study metastability in spin glasses (section 2). They then proceed to evaluate the Rényi complexity of two prototypical disordered systems, the random free energy model (section 3) and the spherical p-spin model (section 4). They obtain consistent predictions for the Rényi complexity, identifying 3 phases: a low temperature phase $T<T_K$ where the complexity is 0, an high temperature phase $T>T_c(m)$ where the complexity is obtained with a replica-symmetric ansatz, and an intermediate regime where the complexity is evaluated with a 1RSB approach.

Given the strengths I listed above, and considering the minor nature of the weaknesses, I consider the paper interesting and suitable for this journal. Therefore, I am willing to recommend it for publication, possibly after addressing the minor changes listed below.

Requested changes

1- The Edwards-Anderson order parameter $q_{\mathrm{EA}}$ is used in Eq. (12) before being defined. In the rest of the paper, the authors define it as the overlap at which the annealed Franz-Parisi potential develops a second minimum below the ($m$-dependent) “Mode-Coupling temperature”. A few lines around Eq. (12), possibly explaining how this definition is consistent with the historical one given by Edwards and Anderson, could help the reader. Notice that, according to the authors’ definition, $q_{\mathrm{EA}}=q_{\mathrm{EA}}(m,T)$, such that the identification with the physical overlap is non-trivial.

2- The behavior of the p-spin model is slightly different than the one of the RFEM, where the Rényi complexity is well defined for all $T$. Indeed, as clear from Fig. 6, there are 3 non trivial temperatures for each index $m$: from lower temperatures, $T_K$ where the complexity starts being non-zero, $T_c(m)$ where it becomes RS, and a generalized mode-coupling transition at a temperature that I will call $T_d(m)$, as it seems to depend on $m$. To clarify this fact, I suggest the authors to define explicitly this temperature in the text, possibly as $T_d(m)$, reserving $T_{\mathrm{MC}}=\lim_{m\to 1 } T_d(m)$.

Recommendation

Publish (meets expectations and criteria for this Journal)

---

## Round 1 · Referee Report · Anonymous (Referee 2) · 2025-2-10

Strengths

1-This is a very pedagogical paper and clearly written
2-The results are robust and reproducible

Weaknesses

1-The motivation for the work is not very clearly stated

Report

I think that this paper can be a valuable addition to the literature, and it can be published on SciPost, but in present form the motivation for the work are not very clearly stated.

My doubt is the following. From the very definition of Renyi complexity, Eq.(16), the link with the Monasson formalism is evident (unless I missed something). In fact one has $p_\alpha = w_\alpha/Z$ where $w_\alpha=\exp[-\beta N f_\alpha]$ is the weight of the state. Then, Eq.(16) gives
$\Sigma \propto \log \sum_\alpha(p_\alpha)^m = \log[ \sum_\alpha w_\alpha^m/Z^m]=\log\sum_\alpha w_\alpha^m - m \log Z = -\beta [\phi(m) -\phi(1)]$, where $\phi(m)$ is the Monasson free energy, as given in Eq.(26) of the paper.

Given that the two quantities are mathematically equivalent, what is the added value of the Renyi entropy with respect to the Monasson calculation? Since from the theoretical point of view the two quantities are equivalent, the only motivation for considering the Renyi entropy must come from the better accessibility of this quantity in numerical calculations. The authors mention this in the introduction, but in my opinion they could make a stronger point.

For example, could it be possible to make a numerical simulation, even in a very simple model, to show that computing the Renyi entropy is easier than computing the Monasson free energy or the Franz-Parisi potential in the same model? This would really bring an added value to the paper.

If this is not possible, I would at least encourage the authors to make a stronger case in favor of the use of the Renyi entropy in the introductions and conclusions.

Recommendation

Ask for minor revision

---

## Editorial Decision

resubmitted